# Extracellular Vesicle-Related Non-Coding RNAs in Hepatocellular Carcinoma: An Overview

**DOI:** 10.3390/cancers16071415

**Published:** 2024-04-04

**Authors:** Giuseppa Augello, Alessandra Cusimano, Melchiorre Cervello, Antonella Cusimano

**Affiliations:** 1Institute for Biomedical Research and Innovation, National Research Council (CNR), 90146 Palermo, Italy; alessandra.cusimano01@unipa.it (A.C.); melchiorre.cervello@irib.cnr.it (M.C.); 2Department of Biological, Chemical and Pharmaceutical Science and Technology (STEBICEF), University of Palermo, 90128 Palermo, Italy

**Keywords:** hepatocellular carcinoma (HCC), non-coding RNAs, extracellular vesicles (EV), micro(mi) RNAs, long non-coding (lnc) RNAs, circular (circ) RNAs, tumor progression, immune escape, drug resistance, biomarkers

## Abstract

**Simple Summary:**

Hepatocellular carcinoma is a highly malignant tumor with increasing prevalence worldwide. Extracellular vesicles exert their biological functions via the delivery of different biomolecules, including non-coding RNAs. In this review, the diverse roles of the non-coding RNA cargo of hepatocellular carcinoma-derived extracellular vesicles will be discussed. Their function in tumor progression, immune escape, and drug resistance, as well as their potentialvalue as biomarkers of disease, will be summarized.

**Abstract:**

Hepatocellular carcinoma (HCC) is the most common primary liver cancer. It is a major public health problem worldwide, and it is often diagnosed at advanced stages, when no effective treatment options are available. Extracellular vesicles (EVs) are nanosized double-layer lipid vesicles containing various biomolecule cargoes, such as lipids, proteins, and nucleic acids. EVs are released from nearly all types of cells and have been shown to play an important role in cell-to-cell communication. In recent years, many studies have investigated the role of EVs in cancer, including HCC. Emerging studies have shown that EVs play primary roles in the development and progression of cancer, modulating tumor growth and metastasis formation. Moreover, it has been observed that non-coding RNAs (ncRNAs) carried by tumor cell-derived EVs promote tumorigenesis, regulating the tumor microenvironment (TME) and playing critical roles in the progression, angiogenesis, metastasis, immune escape, and drug resistance of HCC. EV-related ncRNAs can provide information regarding disease status, thus encompassing a role as biomarkers. In this review, we discuss the main roles of ncRNAs present in HCC-derived EVs, including micro(mi) RNAs, long non-coding (lnc) RNAs, and circular (circ) RNAs, and their potential clinical value as biomarkers and therapeutic targets.

## 1. Introduction

Hepatocellular carcinoma (HCC) is a tumor with a typically poor prognosis [1]. Currently, it is the fourth main cause for cancer-associated death and has the sixth highest incidence of malignancy in the world, with 780,000 deaths and 1 million new cases per year, as of 2021 [1,2,3]. HCC is treated via hepatic resection only in its early stage, and patients with an early diagnosis often have a good prognosis, with a 5-year overall survival (OS) rate of 60–80% [4]. However, the intrahepatic and extrahepatic recurrence rate within 5 years of surgical resection remains high [4]. HCC usually lacks obvious symptoms, and 70% of patients are diagnosed at a late stage, when tumor resection is not possible [5]. HCC detection is based on abdominal ultrasonography (US) and elevated serum α-fetoprotein (AFP) levels [6]. Although abdominal US is highly accurate, its ability to detect small nodules is limited, and measuring AFP levels is suboptimal for early detection due its sensitivity to early-stage tumors being very low [6]. Therefore, there is a need to discover novel markers for liver cancer diagnosis. In patients with advanced stage HCC, the Food and Drug Administration’s (FDA) approved systemic treatment options are sorafenib, lenvatinib, cabozantinib, and immunotherapy (pembrolizumab and nivolumab); however, these treatments only partially improve the outcomes of these patients [7]. Indeed, long-term outcomes are still poor, and therefore new treatment strategies are urgently needed [7].

Extracellular vesicles (EVs) are a heterogeneous population of lipid bilayer-delimited particles naturally released from cells, and they are extensively distributed in various biological fluids [8,9]. Based mainly on their size, EVs can be grouped into exosomes (50–150 nm in diameter), ectosomes (100–1000 nm in diameter), apoptotic bodies (1–5 μm in diameter), and large oncosomes (1–10 μm in diameter) [10]. Exosome formation happens via the endocytosis of the cell membrane forming the early endosomes. Subsequently, early endosomes integrate active molecules and other substances forming multiple intraluminal vesicles (ILVs). Late endosomes mature into multivesicular bodies (MVBs), which can encapsulate cargoes derived from various organelles, including the trans-Golgi network (TGN) and endoplasmic reticulum (ER) compartments [11]. Subsequently, MVBs can fuse with lysosomes [12], and their content becomes degraded; some degradation products, such as amino acids, are made available for reuse by the cell. Furthermore, MVBs are translocated to the membrane and, upon fusion, release vesicles into the extracellular space [13]. In the extracellular space, small- (<200 nm) and medium-/large- (>200 nm) sized vesicles are released, collectively termed “extracellular vesicles”. The EV membranes consist of a variable mixture of lipids and proteins. Besides membrane-inserted proteins, EVs can also contain biomolecules adsorbed onto their surface, also known as protein coronas [14].

Wolf et al. demonstrated that the protein corona on the surface of EVs can also promote normal angiogenesis in vivo [14]. Proteomic analysis highlighted an enrichment of proangiogenic factors in EVs, and the removal of the protein corona from EVs significantly reduced their angiogenic potential [15].

EVs are released by cells carrying a variety of biologically active molecules [16]. EVs can carry lipids, RNA molecules, DNA, soluble proteins (enzymes, cytokines, chemokines, growth factors), and other proteins, such as tumor suppressors, oncoproteins, and transcriptional and splicing regulators [16]. Furthermore, they are also used as “garbage bags” for discharging unwanted molecules [17].

EVs are natural transporters of various nutrients, as well as lipids and proteins, and other components, such as mineral compounds, which are necessary for tissue generation can be observed within them. In particular, osteoblasts shed EVs known as matrix vesicles [18] with enhanced ASMTL-AS1 expression, containing phosphatases, calcium, and inorganic phosphate, which are fundamental for hard tissue maturation [18]. These EVs play an important role in matrix mineralization; however, on the other hand, ectopic matrix vesicles pathologically promote undesirable calcification in vascular tissues [19].

EVs also carry a variety of regulatory proteins, such as proteins that remodel the extracellular matrix (ECM) [20], as well as mediate the intercellular signal transmission [21]. EVs play an important role in controlling the tumor microenvironment (TME), and they can control the development, immune escape, angiogenesis, invasion, and migration of certain cancers [22].

The isolation of EVs is a critical process, with the chosen method significantly impacting both the sample yield and purity [23,24]. The challenge in EV isolation stems not only from their nano-size, but also from potential contaminants, such as cellular debris, lipoproteins, protein complexes, albumin, argonaute protein complexes, vault viral particles, and aggregates carrying nucleic acids, which can be co-isolated with EVs [23,24,25,26,27].

Among the various protocols available, ultracentrifugation protocols, employing differential sequential cycles at 4 °C with forces up to 120,000 g, are widely used for EV isolation. However, EVs isolated through ultracentrifugation may exhibit impaired functionality or form aggregates due to the forces exerted during high-speed centrifugation [28,29,30].

Another common method for EV isolation based on size is ultrafiltration, using membrane filters with specific size-exclusion limits [31,32,33,34]. However, preparations obtained via ultrafiltration often contain contaminants of similar diameters to EVs [31,32,33,34].

Size-exclusion chromatography is gaining popularity for EV isolation via fractionation, which yields high-purity preparations. This method filters samples through a porous stationary phase, based on hydrodynamic radii [29,35,36,37,38].

An alternative is precipitation methods, in which vesicle aggregates are formed by the addition of water-excluding polymers like polyethylene glycol (PEG) [39,40]. However, there is a risk of co-precipitating non-EV material. Commercial kits utilizing this method often include pre-isolation and post-isolation steps in order to minimize contamination [41,42,43,44].

Immunoaffinity techniques, exploiting surface proteins and receptors, have been developed to complement other isolation methods [45,46]. These techniques enhance efficiency, specificity, and integrity in recovering EVs from complex fluids, although challenges may arise from the antibody availability and marker presence across the entire EV population [47].

Recent advancements include isolation methods based on microfluidic technologies, which introduce innovative approaches to EV isolation [48,49].

After isolation, the second step in the study of EVs is their characterization in terms of physical and biochemical characteristics, as well as content [27]. The most common approach to defining the physicochemical and molecular characteristics of EVs is single-particle analysis [27,50]. The main single-particle analysis techniques used in the field of EVs are nanoparticle tracking analysis (NTA), electron microscopy, cryo-electron microscopy, atomic force microscopy, high-resolution microscopy, high-resolution flow cytometry, and Raman spectroscopy [50]. Such basic methods as protein quantification, Western Blot, qPCR, and omics analysis, including proteomics, lipidomic analysis, and RNA sequencing, are the main methodologies used to characterize EV cargoes [27].

EV secretion is an emerging mechanism by which tumor cells communicate with their surrounding environment [51]. Moreover, in cancer, EVs can transfer a variety of molecular factors to the target organ, inducing the inflammation necessary to create pre-metastatic niches [52]. In recent years, EVs have been increasingly recognized as important cancer biomarkers because of their high concentration in blood and other body fluids, and because of the bioinformation they carry from their cells of origin in the form of genes and proteins [53]. Compared to normal cells, cancer cells produce more EVs, and the number in the blood circulation of cancer patients is higher than the number in healthy people [53].

Non-coding RNA (ncRNA) is a nucleic acid that can be packaged within EVs and transferred among tumor cells [54]. The peculiar structure of EVs, with bilayer membranes, supports ncRNA transmission and protects them from degradation via circulatory nucleases. The most common types of ncRNAs are microRNA (miRNA), long non-coding RNA (lncRNA), and circular RNA (circRNA) [54]. In recent years, many reports have shown that EV-related ncRNAs play a pivotal role in many biological and pathological processes [55]. Studies regarding EV-related ncRNAs derived from HCC cells have been increasing, indicating that further knowledge of EV-related ncRNAs in liver cancer is significant for early diagnosis and treatment. In this review, the latest studies on various EV-related ncRNAs in HCC will be discussed, with a focus on their potential value as biomarkers of disease, as well as in tumor progression, drug resistance, and immune escape (Figure 1).

Figure 1 demonstrates some of the potential roles and applications that, to date, have been described in the literature following the analysis of the expression of different types of ncRNAs (miRNAs, lncRNAs, and circRNAs) in patients with HCC, or also in cell and animal models of this tumor.

## 2. miRNA-Extracellular Vesicles

miRNAs are small, non-coding, single-stranded RNA molecules, 19–25 nucleotides long, which have been implicated in the regulation of post-transcriptional gene expression via silencing messenger RNAs (mRNAs) [56]. miRNAs bind the 3′ untranslated region (3′ UTR) of the target mRNA, either blocking its translation or degrading it [56]. miRNAs play a crucial role in cell proliferation, differentiation, angiogenesis, metastasis, chemo-resistance to anticancer drugs, malignancy, and apoptosis; however, the information regarding these small non-coding RNAs in HCC-derived EVs is still limited [56,57] (Table 1).

Furthermore, miRNAs contained in HCC-derived EVs are found in biological fluids, meaning they can also be considered cancer biomarker candidates [58]. Different EV cargoes, including miRNAs, are associated with different tumor stages, allowing the early and late stages of the disease to be distinguished, highlighting their potentially important potential diagnostic and prognostic role [59].

### 2.1. miRNA-Extracellular Vesicles and Hepatocellular Carcinoma Progression

EVs are crucial mediators of both autocrine and paracrine cell communications among different types of liver cells (stellate cells, hepatocytes, and immune cells, including T and B cells, natural killer cells, and Kupffer cells), imperative for maintaining a physiological state. In a pathological state such as cancer, aberrantly expressed miRNAs contained in EVs may act as oncogenes in the absence of cell cycle control, similarly to in HCC, or they may act as tumor suppressors if miRNA blocks the expression of genes involved in cell proliferation [60,61,62].

Tang et al. found lower levels of exosomal miR-9-3p in the serum of HCC patients compared to control sera. Indeed, miR-9-3p binds and blocks HBGF-5 mRNA expression, which has been implicated in cell proliferation, as demonstrated by their subsequent studies in human HCC cell lines (SMMC-7721, HepG2, and QGY-7703). Furthermore, miR-9-3p decreases the expression of the ERK1/2 regulating cell cycle, cell proliferation, and cell development [62].

Conversely, as demonstrated by Cabiati et al. [57], EVs with high levels of miR-27a are more secreted by human HCC HepG2 cells than non-tumor hepatocytes. miR-27a promotes human HCC cell (HepG2, Bel-7402, and Bel-7404) proliferation through PPAR-γ suppression; in fact, miRNA overexpression is closely related to PPAR-γ downregulation. PPAR-γ activation arrests the cell cycle via p21 and p53, as well as inducing apoptosis activating Fas [61].

High exosomal miR-429 expression in human HCC cells is involved in HCC progression, targeting Rb binding protein 4 (RBBP4) and activating E2F1, thus promoting POU class 5 homeobox 1 (POU5F1) expression [63,64].

SMMC-7721 cells secrete EVs containing miR-221, which targets the p27/Kip1 tumor suppressor gene, thus promoting tumor cell proliferation and migration [65].

The expression levels of exosomal miR-665 in the sera of 30 HCC patients were significantly increased when compared with a control group of healthy subjects. Indeed, Qu et al. demonstrated in vitro that miR-665 contained in human MHCC-97H- and MHCC-97L cell-derived EVs promotes cell proliferation and tumor growth via the activation of the MAPK/ERK pathway [66].

Zhou et al. indicated that HCC-derived exosomal miR-21 isolated from 97H and LM3 human liver cancer cell lines could convert normal HSCs (hepatic stellate cells) into CAFs (cancer-associated fibroblasts) by decreasing the PTEN expression, leading to the activation of the PDK1/AKT pathway. CAFs also promoted cancer progression, increasing the secretion of matrix metalloproteinase 2 (MMP2), MMP9, vascular endothelial growth factor (VEGF), tumor growth factor-β (TGF-β), and basic fibroblast growth factor (bFGF) [67].

miR-93 from human SKHEP1 and HuH7 cell-derived EVs promotes HCC proliferation and invasion via directly inhibiting TIMP2/TP53INP1/CDKN1A. Furthermore, the overexpression of miR-93 leads to poor prognosis in HCC patients [68].

Yang et al. demonstrated that human MHCC-97H cell-derived EVs with miR-3129 promoted HCC proliferation and epithelial-mesenchymal transition (EMT) in vitro, and they also promoted HCC growth and metastasis formation in vivo [69].

### 2.2. miRNA-Extracellular Vesicles and Immune Escape in Hepatocellular Carcinoma

In addition to tumor progression, HCC-derived EVs modulate many immune cells, thereby attenuating the anti-HCC immune response [60].

Macrophages play an important role in tumors because they participate in innate immunity. Tumor-associated macrophages (TAMs) consist of two subgroups, M1 and M2 [70]. M1 phenotype macrophages expressing pro-inflammatory cytokines inhibit cell proliferation, whereas the M2 phenotype produces anti-inflammatory cytokines, including TGF-β and IL-10, and demonstrates immunosuppressive and pro-tumoral effects [70,71].

In this context, Liu et al. found that human Hep3B and HepG2 cells transmit exosomal-miR-23a-3p to M2 macrophages, inhibiting PTEN expression and activating the PI3K/AKT signaling pathway; thus, PD-L1 expression increases, and T cell activities are inhibited. Therefore, miR-23a-3p performs a key role in tumor cell escapes from immune cytotoxicity [72].

miR-146a-5p, overexpressed in mouse Hepa1-6 cell-derived EVs, promotes the downregulation of TNF-α and the differentiation of CD206+ macrophages [73]. MiR-452-5p, secreted by human SNU-182 or Huh-7 cell-derived EVs, induces the polarization of M2 macrophages, accelerating HCC growth and metastasis by targeting the tumor suppressor TIMP3 [74]. Similarly, human HepG2 and HuH7 cell-derived exosomal miR-21-5p promotes M2 polarization, enhancing the production of pro-tumorous cytokines. Indeed, miR-21-5p is associated with poor prognosis in HCC [75].

Murine HCC H22 cells release EVs loaded with miRNA let-7b, which binds macrophages, blocking IL-6 production. Thus, this miRNA attenuates tumor inflammation [76].

miR-92b loaded in Hep3B cell-derived EVs downregulates CD69 expression in NK92 cells, influencing its activity against Hep3B cells [77].

### 2.3. miRNA-Extracellular Vesicles and Hepatocellular Carcinoma Drug Resistance

Drug resistance, occurring when cancer cells develop resistance to drugs, is the primary cause of chemotherapy failure [78,79]. The main mechanisms of resistance are the increased transport of efflux pumps, such as P-glycoprotein, the overexpression of multidrug resistance proteins, decreased drug uptake, increased resistance to apoptosis, and changes in drug target levels [78].

Sorafenib is an antineoplastic agent used to treat patients with advanced HCC [80]. It acts by inhibiting RAF kinases, the platelet-derived growth factor receptor (PDGFR-β), the vascular endothelial growth factor (VEGFR), and several other kinases [80]. Moreover, sorafenib may induce p53 expression, causing Forkhead box M1 (FOXM1) suppression. miR-25 in HepG2- and Huh7 cell-derived EVs blocks p53 expression in cancer cells, inducing the expression of FOXM1 and activating the HGF/Ras pathway. This results in the ineffective treatment of HCC with sorafenib [81].

Wang et al. [82] found that exosome-carried miR-744 l was decreased in both HepG2 cells when compared to LO2 cells, and in exosomes from the serum of HCC patients when compared with those derived from healthy individuals. These authors found that PAX2, highly expressed in HCC tissues compared to normal tissues, is a target of miR-744. When miR-744 is downregulated, the expression of PAX2 increases in HCC cells, promoting proliferation and sorafenib chemoresistance [82].

Human HCC cells (CSQT-2 and HCC-LM3) secrete exosomal miR-1247-3p, which decreases β-1,4-galactosyltransferase III (B4GALT3) expression, activating the β1-integrin/NF-κB pathway in fibroblasts that become CAFs [83]. CAFs activated by miR-1247-3p increase their secretion of IL-6 and IL-8, and other pro-inflammatory cytokines, thus leading to stemness, EMT, and chemoresistance to sorafenib [83,84].

### 2.4. miRNA-Extracellular Vesicles as Biomarkers in Hepatocellular Carcinoma

New non-invasive biomarkers, such as EV-loaded miRNAs, have been attracting the attention of the scientific community for HCC diagnosis, prognosis, and the prediction of therapy response [85,86].

There are several studies on EVs containing ncRNA as cancer biomarkers, but only recently has research on miRNAs as HCC biomarkers increased [86]. Many miRNAs are recognized for this role, such as miR-224, which is expressed at a significantly higher level in HepG2 and SKHEP1 cells than in healthy liver cells [87]. Similarly, serum exosomal miR-224 expression levels are higher in patients with HCC than in healthy controls. Furthermore, high exosomal expression of miR-224 in the serum of HCC patients is linked to large and advanced stage tumors [87].

miR-21-5p is upregulated more in plasma-derived exosomes from patients with HCC compared to patients with liver cirrhosis; instead, miR-92a3p is downregulated [88]. Moreover, Sorop et al. proposed a statistical model that integrates these two miRNAs with AFP as a novel screening tool to differentiate HCC from liver cirrhosis [88].

Cho et al. identified miR-4661-5p in HCC-derived EVs as a reliable biomarker of HCC at all stages [89]. The authors found that exosomal miR-4661-5p is a more effective and accurate serum biomarker for HCC diagnosis than serum AFP alone [89].

When analyzing the sera of 178 subjects, which included 28 healthy individuals, 27 patients with CHB (chronic hepatitis B), 33 patients with LC (liver cirrhosis), and 90 patients with HCC, exosomal miR-10b-5p was only present in high levels in HCC patients; therefore, it is considered a potential serum biomarker specifically in the early stages of HCC [90].

Exosomal miR-655, described above, may be considered a new biomarker for HCC diagnosis and prognosis [66].

EV-carried miR-3129 is more highly expressed in the plasma of HCC patients than in healthy subjects, and it is considered a potential new tool for HCC diagnosis [69].

Boonkawe et al. demonstrated that plasma-derived EV-miR-19-3p was significantly elevated in patients with HCC when compared to healthy individuals [91]. Therefore, this finding may allow exosomal miR-19-3p to be used as a valid biomarker for the early diagnosis of HCC [91].

High levels of miR-23a are associated with poor HCC prognosis. In fact, decreased levels of miR-23a block cell proliferation, as demonstrated by Bao et al. [92]. Cabiati et al. also demonstrated that exosomal miR-23a, secreted by human HepG2 cells, is increased in HCC patients when compared to controls; therefore, it is used as a biomarker for the diagnosis of the early stages of HCC [57].

Finding HCC-derived EV-loaded miRNAs for use as new biomarkers will improve the performance of HCC surveillance systems and treatment in clinical practice. This is because several studies have demonstrated the great potential role of EV-miRNAs in cancer diagnosis, prognosis, tumor staging, and the prediction of therapy response [86].

**Table 1 cancers-16-01415-t001:** The biological functions and gene targets of exosomal miRNAs in HCC.

mRNA	Gene Target	Function	Type of Extraction	Reference
miR-9-3p	HBGF-5	cell proliferation	ultracentrifugation	[62]
miR-27°	PPAR-γ	cell proliferation	ultracentrifugation	[61]
miR-429	RBBP4	cell formation/progression	ultracentrifugation	[63]
miR-221	p27/Kip1	cell proliferation/development	ultracentrifugation	[65]
miR-665	MAPK/ERK	cell proliferation/biomarker	precipitation	[66]
miR-21	PTEN	cell proliferation/progression	ultracentrifugation	[67]
miR-93	TIMP2/TP53INP1/ CDKN1A	cell proliferation/invasion	precipitation	[68]
miR-3129	TXNIP	cell proliferation/ EMT/biomarker	ultracentrifugation	[69]
miR-23a-3p	PTEN	immune escape	-	[72]
miR-146a-5p	-	immune escape	ultracentrifugation	[73]
miR-452-5p	TIMP3	immune escape	gradient centrifugation	[74]
miR-21-5p	RhoB	immune escape/biomarker	ultrafiltration/ ultracentrifugation/ precipitation	[75,88]
miRNA let-7b	IL-6	immune escape	ultracentrifugation	[76]
miR-92b	CD69	immune escape/angiogenesis/metastasis	precipitation	[77]
miR-25	p53	drug resistance	ultracentrifugation	[81]
miR-744	PAX2	cell proliferation/drug resistance	ultracentrifugation	[82]
miR-1247-3p	B4GALT3	cell proliferation/EMT/drug resistance	ultracentrifugation	[83]
miR-224	GNMT	cell proliferation/biomarker	precipitation	[87]
miR-92°-3p	PTEN	cell proliferation/EMT/metastasis/ biomarker	precipitation	[88]
miR-4661-5p	IL10	biomarker	precipitation	[89]
miR-10b-5p	-	biomarker	ultracentrifugation	[90]
miR-19-3-p	-	biomarker	precipitation	[91]
miR-23a	-	cell proliferation/biomarker	ultracentrifugation	[92]

## 3. lncRNA-Extracellular Vesicles

About 75% of the human genome is actively transcribed, but only 2% of this is represented by protein-encoding sequences [93]; the other sequences left are ncRNAs, which are RNAs that lack protein-coding capacity. Belonging to this latter group, lncRNAs are ncRNAs with a length of more than 200 nucleotides [94]. It has been demonstrated that lncRNAs play a pivotal role in various physiological and pathological processes like cancers, including HCC [95] (Table 2).

So far, more then 50,000 genes have been identified that transcribe lncRNAs [96]. The expression levels of lncRNAs vary depending on tissue type, and their expression is deregulated in pathological conditions [97]. lncRNAs are important intracellular regulators which can regulate gene expression via interacting with DNA binding promoters or distal regulatory elements and recruiting epigenetic modifiers [94].

lncRNAs can regulate mRNA stability by binding it. In addition, lncRNAs can also bind to miRNAs like molecular sponges, thus interfering with their activities.

lncRNAs have the ability to engage with proteins, contributing to the assembly of protein complexes, where they serve as a scaffold. Additionally, some lncRNAs have the capacity to encode small peptides, through which they exhibit their biological functions. lncRNAs can be packaged into exosomes to protect them from degradation [98].

### 3.1. lncRNA-Extracellular Vesicles and Hepatocellular Carcinoma Tumor Progression

The analysis of the autocrine effects of the EV contents released by tumor cells has highlighted the role of numerous EV-enriched lncRNAs.

lncRNA FAL1 improves liver cancer cell proliferation and migration by binding to miR-1236, which is a regulator of hypoxia-induced EMT and a cell metastasis biomarker [99].

lncRNA FAM138B (linc-FAM138B) is downregulated in HCC, and its low expression correlates with a poor prognosis [100]. linc-FAM138B can be packaged in exosomes (exo-FAM138B) and released by HCC cells. HCC cells treated with exo-FAM138B show reduced proliferation, migration, and invasion due to the inhibition of miR-765 levels [100]. In vivo data from a xenograft model confirmed that linc-Fam138B-EV suppresses tumor growth [100].

Plasma exosomal lncRNA RP11-85G21.1 (lnc85) stimulated HCC cellular proliferation and migration by targeting miR-324-5p [101].

Serum exosomal LINC00161 has been recognized as a potential biomarker for HCC [102]; however, in vitro exosomes derived from human HCC cells were able to stimulate the proliferation, migration, and angiogenesis of human umbilical vein endothelial (HUVEC) cells. In vivo experiments in mice showed that LINC00161 induces tumorigenesis and metastasis in HCC by targeting miR-590-3p and, consequently, activating its target, ROCK2 [102].

lncRNA ASMTL-AS1 is highly expressed in human HCC tissues and delivered by exosomes [103]. Enhanced ASMTL-AS1 expression induces cell proliferation and migration, along with promoting invasion and EMT in Huh7 cells via ASMTL-AS1/miR-342-3p/NLK/YAP axis [103].

Treatment with exosome-depleted lncH19 from human Huh7 cells inhibited proliferation, migration, and invasion, and induced apoptosis, in HCC cells, whereas exosomal lncH19 from untreated Huh7 cells promoted the proliferation and metastasis, and inhibited the apoptosis, of HCC cells [104]. Moreover, the authors found that the overexpression of miR-520a-3p reversed the effects of treatment with H19-Propofol-Huh7-exo [104]. These data, therefore, suggested that high levels of exosomal lncH19 enhanced tumor development in HCC sponging miR-520a-3p [104].

The lncRNA muskelin 1 antisense RNA (MKLN1-AS) has been demonstrated to play a growth-promoting role in HCC [105].

Special environmental conditions, such as hypoxia, can modulate the release of EVs and define specific cargos [106]. This is the case for linc-ROR, which is induced and enriched in exosomes produced by HCC cells under hypoxic stress [106]. Linc-ROR in recipient cells is able to induce the expression of hypoxia-inducible factor 1 α (HIF1a) and its downstream targets, such as pyruvate dehydrogenase kinase isozyme 1 (PDK1) [106].

lncRNA TUG1 presented in human CAF-secreted exosomes promoted migration, invasion, and glycolysis in human HepG2 cells [107].

In a model of DEN-induced HCC in rats, treatment with hepatic cancer stem cell (CSC)-derived exosomes upregulated the exosomal lncHEIH, lncHOTAIR, and lncTuc339 in rat liver cancer cells [108]. Conversely, the expression of these three lncRNAs was downregulated in livers treated with mesenchymal stem cell (MSC) exosomes derived from rat bone marrow [108]. Therefore, exosomes released from the two different stem cells have opposite effects on HCC: pro-tumoral for CSC-exosomes, and anti-tumoral for MSC-exosomes [108]. All three lncRNAs are involved in the regulation of cell proliferation and metastasis [109], and lncHOTAIR played a role in the transformation of normal liver stem cells into CSCs [110].

### 3.2. lncRNA-Extracellular Vesicles as Biomarkers in Hepatocellular Carcinoma

Numerous exosome-contained lncRNAs have been the subject of extensive research for their considerable diagnostic value as potential and promising liquid biopsy markers [98].

From the in silico analysis of human liver cancer tissue datasets, Su et al. identified a predictive signature based on five upregulated exosome-related lncRNAs (AC099850.3, LINC01138, MKLN1-AS, AL031985.3, and TMCC1-AS1) which have been associated with poor prognosis in liver cancer patients [111]. In the risk model created by the authors, the signature is correlated with the elevated expression of some crucial regulatory mechanisms of the immune system, and is related to the tumor immune microenvironment [111].

Kim et al. identified four serum EV-derived lncRNAs, SNHG1, MALAT1, HOTTIP, and DLEU2, which were significantly elevated in HCC patients when compared to non-HCC patients, and which significantly discriminated between the two groups [112].

Exosomal LINC00161 has been found to be upregulated in serum exosomes of HCC patients when compared to healthy subjects, and, as suggested by the authors, can be a valuable marker for the diagnosis of HCC [113].

The exosomal lncSENP3-EIF4A1 exhibits reduced expression in HCC patients compared to healthy controls [11]. Both in vitro and in vivo studies have demonstrated that the introduction of SENP3-EIF4A1 functions as a tumor suppressor, inhibiting the proliferation and migration of HCC cells and acting as a competitor of the miR-9-5p target [114].

Serum exosomal lncRNA CRNDE expression levels were shown to be higher in patients with HCC when compared to normal controls [115]. In addition, high levels of serum exosomal lncRNA CRNDE were reported to be correlated with poor prognosis in HCC patients [115]. In vitro, the inhibition of CRNDE expression in human HCC cells was associated with reduced cell proliferation, migration, and invasion [116].

Recently, Yao et al. demonstrated that plasmatic exosomal levels of PRKACA-202, H19-204, and THEMIS2-211 were upregulated in HCC patients when compared to normal controls [117]. In this study, the diagnostic value of the combination of exosomal levels of THEMIS2-211 and PRKACA-202 exceeded that of AFP for the diagnosis of early-stage HCC patients. Furthermore, at the molecular level, the authors demonstrated that THEMIS2-211, through its interaction with miR-940, has an oncogenic role in HCC and regulates proliferation, migration, invasion, and EMT via the modulation of the proteoglycan SPOK1 [117].

The exosome-derived lncRNAs CTD-2116N20.1 and RP11-136I14.5 have been recognized as potential biomarkers for predicting the survival rate of HCC patients [118]. Their presence is associated with an unfavorable prognosis in individuals with HCC. CTD-2116N20.1 is suggested to regulate proteins involved in cell proliferation and tumor metastasis [118].

The exosomal lncRNA RP11-583F2.2 was first identified via bioinformatic analysis and then validated in the serum of HCC patients [119]. Its expression is upregulated in HCC patients compared to viral hepatitis C patients and healthy people [119].

Similarly, exosomal lncRNA-HEIH expression levels allowed researchers to distinguish between chronic hepatitis C and hepatitis C virus (HCV)-associated HCC patients, as its expression only increased in exosomes from patients with HCV-related HCC [120].

Plasma exosomal lncRNA RP11-85G21.1 (lnc85) allowed AFP-negative HCC to be distinguished from liver cirrhosis patients and healthy controls [101].

The exosomal lncRNAs ENST00000440688.1, ENST00000457302.2, and ENSG00000248932.1 are differentially expressed in patients with metastatic HCC when compared to those with non-metastatic HCC [121]. LINC00635 and lncRNA ENSG00000258332.1 (LINC02394) were upregulated in HCC patients and, when combined with AFP, were able to distinguish HCC patients in an independent test of validation [122].

The expression of serum exosomal lncRNA-ATB, associated with exosomal miRNA-21, was inversely correlated with the overall survival and the progression-free survival rates of HCC patients [123].

The analysis of EVs derived from the serum of healthy individuals and individuals with hepatitis, cirrhosis, and HCC revealed the differential expressions of lncZEB2-19, lnc-GPR89B-15, lnc-EPC1-4, and lnc-FAM72D-3 among groups [124]. This functional study highlights that lnc-FAM72D-3 acts as an oncogene targeting hsa-miR-5787 expression, while lnc-EPC1-4 functions as a tumor suppressor gene targeting the influx transporters SLCO1B1v and miR-29b-1-5p [124].

### 3.3. lncRNA-Extracellular Vesicles and Immune Escape

Human PLC/PRF/5 and Hep3B HCC cell-derived exosomes were found to be enriched with lncTUC339, which is involved in modulating tumor cell growth and adhesion [109]. Exosome-derived TUC339 serves as a mechanism by which tumor cells influence and modify their surrounding environment; indeed, it can promote tumor growth [109] and modulate macrophage activation [125]. This study demonstrated that exosomes from HCC cells could be taken up by THP-1 cells, where TUC339 induced macrophage M1/M2 polarization, switching from a proinflammatory (M1) to an anti-inflammatory phenotype (M2). In addition, elevated levels of TUC339 were found in M(IL-4) macrophages [125].

The immunomodulatory capacity of HCC-released exosomal lncRNA has been demonstrated by other studies [126]. In HCC, the overexpression of PD ligands (PD-L1 and PD-L2) was linked to poor prognosis in HCC patients [127]. Fan et al. demonstrated that human HepG2 and Huh7 HCC cell lines released lncRNA PCED1B-AS1-containing exosomes, which enhanced PD-L expression in receipt HCC cells. PCED1B-AS1 induced PD-L expression via sponging hsa-mir-194-5p, which inhibited PD-L expression [126].

### 3.4. lncRNA-Extracellular Vesicles and Drug Resistance in Hepatocellular Carcinoma

lncRNA-ROR was found to be enriched in human HepG2 and PLC/PRF/5 HCC cell-derived exosomes, and seemed to be involved in the mechanisms of sorafenib resistance; indeed, lncRNA-ROR levels increased during sorafenib treatment, and inhibited sorafenib-induced cell death [128]. In the same way, lncRNA-VLDLR was involved in the resistance to drugs, such as camptothecin and doxorubicin, as well as sorafenib [129].

**Table 2 cancers-16-01415-t002:** The biological functions and gene targets of exosomal long non-coding (lnc) RNAs in HCC.

lncRNA	Gene Target	Function	Type of Extraction	Reference
lncRNA FAL1	miR-1236	cell proliferation/migration	precipitation	[99]
lncRNA FAM138B	miR-765	cell proliferation/migration/invasion	gradient centrifugation	[100]
lncRNA RP11-85G21.1 (lnc85)	miR-324-5p	cell proliferation/migration	precipitation	[101]
LINC00161	miR-590-3p	tumorigenesis/metastasis	precipitation	[102]
lncRNA ASMTL-AS1	miR-342-3p	cell proliferation/migration/invasion EMT	precipitation	[103]
lncH19	miR-520a-3p	tumor development	precipitation	[104]
lncRNA MKLN1-AS	miR-22-3p	cell growth/angiogenesis/ migration/	precipitation	[105,120]
Linc-ROR	HIF	regulate hypoxia condition/ drug resistance	ultracentrifugation	[106]
lncRNA TUG1	PTEN	migration/invasion/glycolysis	ultracentrifugation	[107]
lnc HEIH		cell proliferation/metastasis/biomarker	ultracentrifugation and density gradient separation	[108,109,120]
lncTuc339		cell proliferation/metastasis Immune escape	ultracentrifugation and density gradient separation	[108,109]
Lnc HOTAIR		cell proliferation/metastasis	ultracentrifugation and density gradient separation	[108,110]
TMCC1-AS1, AL031985.3, LINC01138, AC099850.3		biomarkers	precipitation	[111]
DLEU2, HOTTIP, MALAT1, SNHG1		biomarkers	precipitation	[112]
lncSENP3-EIF4A1	miR-9-5p	immune escape	precipitation	[114]
lncRNA CRNDE		cell proliferation/migration/invasion biomarker	precipitation	[115]
THEMIS2-211	miR-940, modulation of proteoglycan SPOK1	proliferation/migration/invasion/EMT/biomarker	ultracentrifugation	[117]
PRKACA-202		biomarker	ultracentrifugation	[117]
CTD-2116N20.1		biomarker cell proliferation/tumor metastasis		[118]
RP11-136I14.5		biomarker		[118]
RP11-583F2.2	miR-1298	biomarker	precipitation	[119]
ENSG00000248932.1, ENST00000440688.1, ENST00000457302.2		biomarkers	precipitation	[121]
ENSG00000258332.1 (LINC02394), LINC00635		biomarkers	precipitation	[122]
lncRNA-ATB		biomarkers	precipitation	[123]
lnc-FAM72D-3, lnc-GPR89B-15, lncZEB2-19	Has-miR-5787	biomarkers/oncogene	ultracentrifugation	[124]
lnc-EPC1-4	SLCO1B1v and miR-29b-1-5p	biomarker/tumor suppressor	ultracentrifugation	[124]
lncRNA-VLDLR		drug resistance	ultracentrifugation	[129]

## 4. circRNA-Extracellular Vesicles

Circular RNAs (circRNAs) are endogenous non-coding products [130]. They were initially considered as a byproduct of splicing errors, but, thanks to a sequencing approach, it has been found that circRNAs are related to numerous diseases, including tumors [130,131]. circRNA has a closed circular structure, which is more stable and not easily degradable by exonuclease [132]. Moreover, being carried inside exosomes renders them difficult to be degraded by enzymes [133]. In the last few years, the role of circRNAs has been investigated; they play a vital role in biological functions, they are certainly known to function as competing endogenous RNA (ceRNA), and, together with miRNAs, they regulate the stability of target RNAs and gene expression [134,135]. Moreover, it was found that circRNAs, by interacting with proteins, can also function as protein decoys, scaffolds, and recruiters [136], with regulatory roles in different biological processes. Increasing evidence has confirmed the roles of circRNAs in early diagnosis, tumor progression and invasion, immune escape, and drug response in HCC (Table 3).

### 4.1. circRNA-Extracellular Vesicles and Hepatocellular Carcinoma Tumor Progression

Numerous studies have found that exosomal circRNAs may have a dual role, acting as promoter or suppressor of HCC progression, invasion, angiogenesis, metastasis, and EMT [137].

Dai et al. demonstrated that human normal hepatic epithelial (L02) cells exposed to arsenite release circRNA_100284 contained in exosomes [138]. CircRNA_00284 is able to activate the cell cycle and increase the proliferation of normal liver cells by interacting with miR-217 [138].

Exosomal circRNA Cdr1 was reported to be important for HCC cell proliferation and migration. Studies demonstrated that exosomes extracted from human HCC cells have elevated levels of Cdr1, and they can induce the proliferative and migratory capabilities of surrounding normal cells [139].

Yu et al. [140] found that Circ_0061395 induced HCC growth through the miR-877-5p/PIK3R3 pathway. They indicated that Circ_0061395 silencing inhibited HCC cell progression and induced cell cycle arrest and cell death in HCC cells.

Exosomal circTTLL5 was found to be more highly expressed in HCC tissues [141]. At the molecular level, the knockdown of circTTLL5 suppressed HCC cell proliferation in vitro, and, in vivo, inhibited tumor growth in mice [141]. Exosomal circRNAs also mediated the crosstalk between normal and cancer cells, thus regulating human HCC growth and metastasis [142].

EMT is a fundamental biological process for cancer cell invasion [143]. Exosome-circRNAs mediate the EMT of HCC cells. Zhang and colleagues showed that, in human cell lines and in mice xenograft models, circ_0003028, via an exosome pathway, controlled E-cadherin, N-cadherin, and vimentin, and induced the EMT transition [144]. Mechanically, circ_0003028 sponge miR-498, which controlled its downstream target ornithine decarboxylase 1 (ODC1) as well as the knockdown of circ_0003028, inhibited cell proliferation and metastasis, and also promoted apoptosis.

The exosome circWDR25 induced tumor progression and EMT through ALOX15 activation via the miR-4474-3p-sponge in SMMC-7721, Hep3B, and HCCLM3 human cell lines [145].

The overexpression of circ-0004277 enhanced the proliferation, migration, and EMT of HCC cells in nude mice and in human cell lines [146].

Higher levels of circ_002136 were shown to be contained in exosomes from HCC cell lines (HA22T and Huh7) rather than in a normal human liver-derived cell line (THLE-3) [147]. In vitro and nude mice experiments showed that silencing circ_002136 inhibited the growth of human HCC cells via the miR-19a-3p and RAB1A axis [147].

Lyu et al. found that circWHSC1, in addition to being associated with a worse overall survival rate in HCC patients, contributed to HCC growth via the elevation of homeobox A1 (HOXA1) expression through sponging miR-142-3p [148].

### 4.2. circRNA-Extracellular Vesicles and Immune Escape

Wang et al. have shown that circ_0074854 can be transferred from human HCC cells to human macrophages (THP-1 cells) through exosomes [149]. The downregulation of circ_0074854 in exosomes suppressed macrophage M2 polarization, then suppressing the invasion of HCC cells in in vitro and in vivo experiments [149].

Recently, Hu and colleagues reported that a conditioned medium from cultured HCC cells and plasma-exosomes from HCC patients showed high levels of circCCAR1 when compared to a normal human liver cell line and plasma-exosomes from healthy subjects [150]. The same authors have shown that exosomal circCCAR1 was adsorbed by CD8+ T cells, which induced their alteration by inhibiting proliferation, enhancing cell death, lowering cytokines secretion, and, in addition, stabilizing PD-1 protein expression on the cell surface. Thus, circCCAR1 would appear to promote resistance to anti-PD1 immunotherapy [150].

circUHRF1 was overexpressed in HCC cell-derived exosomes and induced NK cell exhaustion [151]. Exosomal circUHRF1 inhibited the secretion of IFN-γ and TNF-α via the induction of TIM-3 expression on NK cells [151]. Furthermore, HCC cell-derived exosomal circUHRF1 may promote the development of resistance to anti-PD1 therapy in a subgroup of HCC patients.

circASAP1, in addition to promoting proliferation and invasion in HCC cells, mediated macrophage infiltration via the miR-326 and miR-532-5p-CSF-1 axis [152].

A recent study found that human HCC cell-derived exosomal circGSE1 promoted the tumor immune escape process by expanding human T cells (Tregs) via the regulation of the miR-324-5p/TGFBR1/Smad3 axis [153]. As a result, the expansion of Tregs promoted the progression of HCC [153].

The ATP-adenosine metabolic pathway regulated by CD39/CD73 is an important immunosuppressive factor in HCC patients; in fact, it induces the culling of NK and T cells [154]. It was demonstrated in both in vitro and in vivo mice models that human macrophages absorb exosome circTMEM181, leading to the regulation of CD39 expression by sponging miR-488-3p [155]. CD39 expression, in synergy with CD73, activates the ATP–adenosine pathway and promotes HCC progression and resistance to anti-PD1 therapy [155]. In contrast, it has recently been shown that circTMEM181 is downregulated in HCC tissues, and decreased expression levels of circTMEM181 were associated with the shorter overall survival of HCC patients [156].

### 4.3. circRNA-Extracellular Vesicles and Drug Resistance in Hepatocellular Carcinoma

circRNAs play a crucial role in the development of drug resistance and radioresistance in different tumors, including HCC [157]. In human HCC, the cell-derived exosomal circRNA-SORE, also named circ_0087293 or circRNA_104797, controlled sorafenib resistance by stabilizing the oncoprotein YBX1 [158]. The research group’s in vitro and in vivo results demonstrated that circRNA-SORE, by binding to YBX1, prevented proteasomal degradation via E3 ubiquitin ligase PRP19 [158].

Hao et al. demonstrated that circPAK1 was overexpressed in the Lenvatinib-resistant human cell lines, LM3 and Hep3B, and that exosomes from resistant cells can mediate the circPAK1 transfer to sensitive cells, thus inducing the lenvatinib resistance of receipt cells [159].

Other authors have demonstrated that circZFR was highly expressed in cisplatin-resistant human HCC cell lines (Huh7 and MHCC97L) and in CAFs [160]. Furthermore, human CAF-derived exosomes delivered circZFR to human HCC cells, inhibited the STAT3/NF-κB axis, and induced cisplatin resistance [160].

Finally, all circRNAs previously described as being involved in immune evasion have the potential to induce resistance to anti-PD1 therapy [156,160,161].

### 4.4. circRNA-Extracellular Vesicles as Biomarkers in Hepatocellular Carcinoma

In earlier works, Sun et al. found that three circRNAs, hsa_circ_0075792, hsa_circ_0004123, and hsa_circ_0004001, were upregulated in the blood samples of HCC patients, and that their expression was positively correlated with TNM stage and tumor size [161].

Other relevant evidence comes from the work of Lin et al., who found that circ_0072088 was highly expressed in human liver tumor tissues, as well as in exosomes isolated from serum of human patients with HCC [162]. Furthermore, its expression was correlated with unfavorable prognosis in HCC patients [162].

CircFBLIM1 has been shown to be upregulated in HCC serum exosomes [163]. CircFBLIM1 was shown to function as an miR-338 sponge [163]. In a mouse model of HCC tumorigenesis, its levels were correlated with HCC glycolysis and tumor progression induced by the circFBLIM1/miR-338/LRP6 axis [163].

Exosomal hsa_circ_0028861 and hsa_circ_0070396 were detected at higher levels in the serum of HCC patients when compared to sera from chronic HBV and cirrhosis individuals [164,165]. Of interest, the integration of hsa_circ_0028861 with AFP showed the best diagnostic capacity in HBV-derived HCC patients than each parameter alone.

In addition, Zhang et al. found that the levels of serum exosomal circTMEM45A were higher in HCC patients when compared to healthy controls, and its level was positively correlated with poor prognosis in patients [166]. In vitro and in vivo experiments demonstrated that circTMEM45A promoted cell mobility and tumorigenesis through the miR-665/IGF2 axis [166].

Recently, circANTXR1 was found at high levels in exosomes from the serum of HCC patients, and showed diagnostic value in distinguishing HCC patients from healthy controls [167]. Furthermore, circANTXR1, acting as an miR-532-5p sponge, promoted the progression of HCC through the upregulation of the DNA double-strand break repair gene XRCC5 [167]. Conversely, it was demonstrated that circ-0051443 contained in HCC tissues was significantly lower when compared to peritumoral tissue and normal tissues, suggesting that exosomal circ-0051443 is a potential biomarker and could be useful for distinguishing HCC tissue from adjacent normal tissues [168]. Furthermore, circ-0051443 was transferred from normal cells to HCC cells, and inhibited tumor progression by blocking the cell cycle and promoting the apoptosis of HCC cells [168].

**Table 3 cancers-16-01415-t003:** The biological functions, gene targets, and extraction types of exosomal circular RNAs in HCC.

circRNA	Gene Target	Function	Type of Extraction	Reference
circRNA_100284	miR-217	cell proliferation	precipitation	[138]
circRNA Cdr1	miR-1270	cell proliferation/invasion	precipitation	[139]
circ_0061395	miR-877	cell proliferation	precipitation	[140]
circTTLL5	miR-136-5p	cell proliferation	precipitation	[141]
circ_0003028	miR-498	cell proliferation/EMT	precipitation	[144]
circWDR25	miR-4474-3p	cell proliferation/EMT	ultracentrifugation	[145]
circ-0004277	ZO-1	cell proliferation/EMT	precipitation	[146]
circ_002136	miR-19a-3p	cell proliferation	precipitation	[147]
circWHSC1	miR-142-3p	cell proliferation	precipitation	[148]
circ_0074854	HuR	cell migration/immune escape	ultracentrifugation	[149]
circCCAR1	miR-127-5p	immune escape	precipitation	[150]
circUHRF1	miR-449c-5p	immune escape	precipitation	[151]
circASAP1	miR-326/ miR-532-5p	cell proliferation/ immune escape	precipitation	[152]
circGSE1	miR-324-5p	cell proliferation/immune scape	ultracentrifugation	[153]
circTMEM181	miR-488-3p	immune escape	ultracentrifugation	[155,156]
circRNA-SORE	miR-660-3p	drug resistance	-	[158]
circPAK1	14–3-3 ζ	cell proliferation/ drug resistance	ultracentrifugation	[159]
circ_0004001 circ_0004123 circ_0075792	-	biomarkers	-	[161]
circ_0072088	-	biomarkers	gradient centrifugation	[162]

## 5. Conclusions

Many studies have confirmed the significant role played by EVs in the pathophysiological processes of HCC. The cargo-based selection carried out in this review (Figure 2) also offers an analysis of the increasingly important role that ncRNAs play in the biology of HCC and the potential utility of these molecules in clinical applications.

EVs and their ncRNA content perform an autocrine action, modulating the cell proliferation and tumorigenicity of the same cell that releases them; they have a paracrine action that conditions the entire tumor environment, modulating the behavior of other cell types, such as stem and immune cells. Finally, they seem to be related to drug resistance, although this aspect has not yet been extensively investigated.

The benefits of studying EVs come from the fact that exosomes originating from tumors contain diverse molecules, including proteins and ncRNA inherited from their parent cells. Consequently, they offer a precise representation of the characteristics exhibited by the tumor cells from which they originated [169]. This makes them crucial players in the fight against cancer, which we need to pay attention to as potential targets for future anti-tumor therapies.

In addition, their presence in biological fluids and the potential for the long-term storage of blood samples containing EVs in a biobank renders them good candidates as biomarkers and a precious tool for the diagnosis and prognosis of the disease. However, the primary obstacle in this context regards the technical challenges associated with the isolation of exosomes, which include a variety of different issues.

Furthermore, although technical challenges currently limit their clinical application, the data collected are promising, underscoring the need for further research in order to improve our understanding of exosomes and their potential significance as cancer biomarkers.

## Figures and Tables

**Figure 1 cancers-16-01415-f001:**
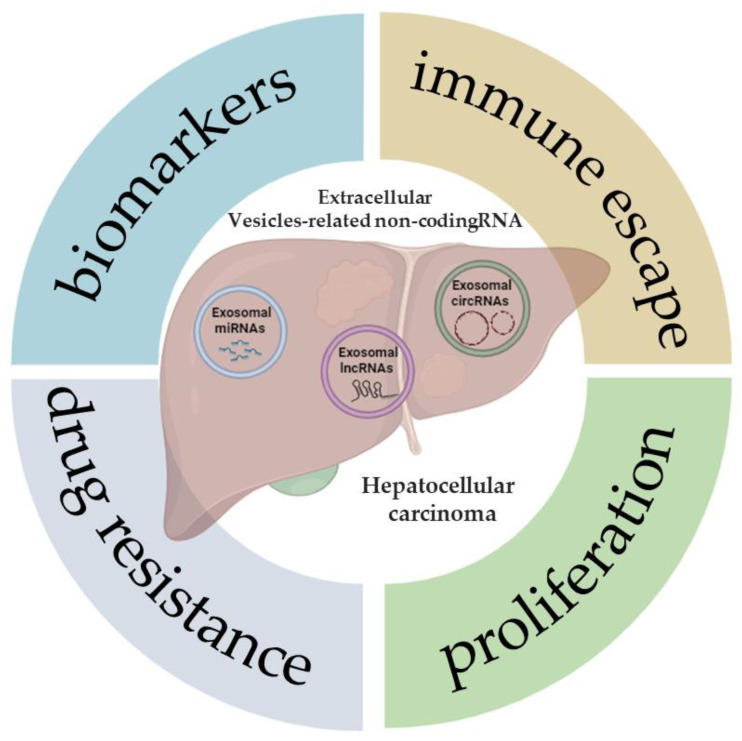
Extracellular vesicle-related non-coding RNAs in hepatocellular carcinoma. The illustration was created with BioRender.com, accessed on 9 January 2024.

**Figure 2 cancers-16-01415-f002:**
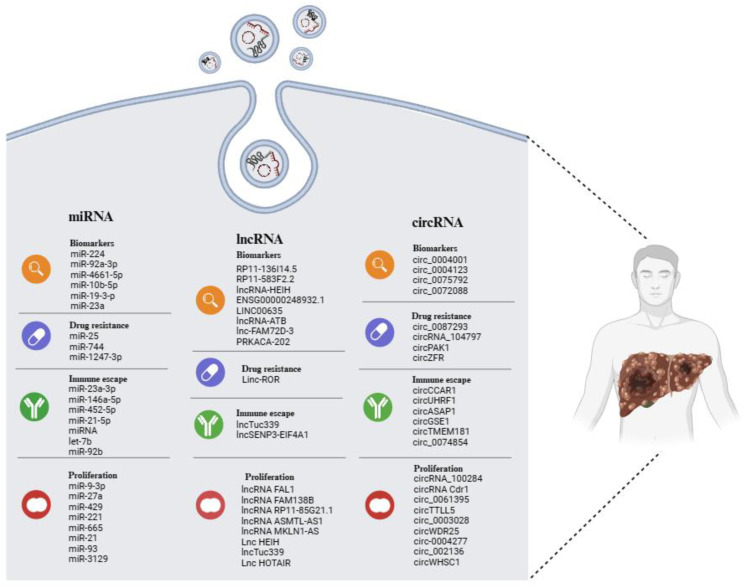
Overview of EV-related ncRNAs in HCC, displaying potential as biomarkers of disease, as well as indicators of tumor progression, drug resistance, and immune escape. The illustration was created with BioRender.com, accessed on 24 February 2024.

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
