# Peer review of "Extracellular Vesicle-Related Non-Coding RNAs in Hepatocellular Carcinoma: An Overview"

_cancers, 2024, doi:10.3390/cancers16071415_

Round 1
Reviewer 1 Report
Comments and Suggestions for Authors
This manuscript gives an overview on extracellular vesicle-derived non-coding RNAs and their role in hepatocellular carcinoma. The authors describe different aspects like the suitability and tissue specificity of miRNAs, lncRNAs and circRNAs in this type of cancer. The review is very well written and gives a comprehensive overview of the state-of-the-art. I just have some minor remarks:
1) I think for the interested reader it would be a great benefit, if there would be a second figure, summarizing the key findings of the research. In addition to the tables, a graphical overview would make it easier for the reader to comprehend the key messages.
2) Working with EVs can be quite challenging. Although the reviewer understands the focus of this overview article, the reader should be made aware of these challenges by mentioning some basics of EV purification and characterization (just one paragraph), also citing the "Minimal Information for studies of extracellular vesicles (MISEV)" guidelines, recently published 2018 and currently in preparation/production in the "Journal of Extracellular Vesicles" (JEV).
Comments on the Quality of English LanguagePlease carefully revise the manuscript in terms of english grammar. e.g. line 76 "EV secretion is an emerging mechanisms....." change to "EV secretion is an emerging mechanism"
Author Response
"Please see the attachment."

Reviewer 2 Report
Comments and Suggestions for Authors
General comments:
I) This is ambitious attempt to review the nanoparticle-related non-coding RNAs in heptacellular carcinoma. However, the review lacked a critical assessment on the findings to grasp the diversity of nanoparticles carrying nucleic acids and their possible effects.
II) One fundamental issue which was not taken sufficiently into account in the review is that extracellular vesicles may not represent the best carriers of nucleic acids. Indeed, several non-membranous nanoparticles, as albumin, argonaute protein complexes, exomeres lipoproteins, supermeres, supramolecular attack particles, vault and viral particles, contain an enriched amount of nucleic acids. Most of the findings associated to extracellular vesicles and nucleic acids need to be re-assessed since the presence of any non-membranous nanoparticles in the samples containing extracellular vesicles can’t be neglected.
III) How extracellular vesicles are extracted and characterized need to be assessed for each of the findings since the properties and characteristics of extracellular vesicles depend on how they are extracted. This needs to be emphasized and critically assessed.
IV) The structure of the review was not well organized. Often findings associated with human body fluids were mixed with cellular findings. Sometimes, the specie origin of the cells was omitted making unclear if it can be adequately extrapolated to other species, including human.
V) Nanoparticles, including extracellular vesicles, are not essentially involved in cell to cell communication. They are also implicated in bringing nutrients, carrying garbage, initiating mineral formation in bone tissues and in pathologic ectopic calcification, just to mention a few functions. This needs to be further commented in the review.
VI) Several sentences were not adequately supported by references. Often it was unclear if the finding reported in the uncited sentence corresponded to the previous sentence or the next one.
VII) Corona particles which can influence properties of the nanoparticles were not discussed.
VIII) Conclusion shall underline the complexity of nanoparticles and their relative variability of the nanoparticule populations which can affect pathophysiological functions.
Minor comments:
1) Title, p1: Replace “Extracellular vesicle” by “Extracellular vesicle and non-membranous nanoparticles”
2) Simple summary, p1: Replace “Extracellular vesicles” by “Extracellular vesicles and non-membranous nanoparticles” in “Extracellular vesicles exert their biological functions by delivering different biomolecules, including non-coding RNAs.”
3) Simple summary, p1: Replace “Extracellular vesicles” by “Extracellular vesicles and non-membranous nanoparticles” in “In this review, the diverse roles of the non-coding RNA cargo of hepatocellular carcinoma-derived extracellular vesicles will be discussed.”
4) Abstract, p1: Add “Non-membranous nanoparticles are also carriers of nucleic acids. Since their presence in the extracellular vesicles can’t be neglected. Extracellular vesicles and nanoparticles shall be considered in the review” or something similar after “Extracellular vesicles (EVs) are nanosized lipid double-layer vesicles containing various biomolecule cargoes, such as lipids, proteins, and nucleic acids.”
5) Abstract, p1: Replace “tumor cell-derived EVs” by “tumor cell-derived EVs and non-membranous nanoparticles” in “Moreover, it has been observed that non-coding RNAs (ncRNAs) carried by tumor cell-derived EVs promote tumorigenesis, regulating the tumor microenvironment (TME) and playing critical roles in the progression, angiogenesis, metastasis, immune escape and drug resistance of HCC.”
6) Abstract, p1: Replace “EV-related ncRNAs” by “EV and non-membranous nanoparticle-related ncRNAs” in “EV-related ncRNAs can provide information on disease status, thus covering a role as biomarkers.”
7) Abstract, p1: Replace “HCC-derived EVs”by “HCC-derived EVs and non-membranous nanoparticles” in “In this review, we discuss the main roles of ncRNAs present in HCC-derived EVs, including micro(mi) RNAs, long non‑coding (lnc) RNAs, and circular (circ) RNAs, and their potential clinical value as biomarkers and therapeutic targets.”
8) Introduction, p1: Add references to support “Hepatocellular carcinoma (HCC) is a tumor with a poor prognosis.
9) Introduction, p1: Add references to support “HCC is treated by hepatic resection only in its early stage, and patients with early diagnosis often have a good prognosis, with a 5-year overall survival (OS) of 60%–80%.”
10) Introduction, p1-2: Add references to support “HCC detection is based on check by abdominal ultrasonography (US) and elevated serum α-fetoprotein (AFP) levels”
11) Introduction, p2: Add references to support “In patients with advanced stage HCC, the Food and Drug Administration’s (FDA) approved systemic treatment options are sorafenib, lenvatinib, cabozantinib and immunotherapy (pembrolizumab and nivolumab), but these therapeutic treatments only partially improve the outcomes of these patients.”
12) Introduction, p2: Indicate that small size and medium size vesicles are often overlapped and that the collective term “extracellular vesicles” is the preferred term after “Furthermore, MVBs are translocated to the membrane, and upon fusion with it, they release vesicles into the extracellular space [13].”
13) Introduction, p2: This is not exactly true. EVs may have other functions, for instance matrix vesicles (a class of EVs) are essentially involved in apatite formation and are less likely to be involved in the cell-cell communication. Besides EVs are implicated as carriers of garbage and again unlikely to be involved in cell to cell communication. Furthermore, the relative proportion of EVs participating in cell to cell communication versus other functions have not been intensively investigated. Rephrase, and add references in “EVs are released by cells carrying a variety of biologically active molecules involved mainly in cell communication.”
14) Introduction, p2: Replace “exosomes” by “extracellular vesicles” in “Exosomes can carry lipids, RNA molecules, DNA, soluble proteins (enzymes, cytokines, chemokines, growth factors), and other proteins, such tumor suppressors, oncoproteins, transcriptional and splicing regulators. [14].”
15) Introduction, p2: Replace “exosomes” by “extracellular vesicles” in “Exosomes play an important role in controlling the tumor microenvironment (TME), and they can control the development, immune escape, angiogenesis, invasion, and migration of certain cancers [17].”
16) Introduction, p2: A paragraph associated to the difficulty of obtaining a pure fraction EVs and that non-membranous nanoparticles can’t be excluded from any preparation of EVs shall be added after “Exosomes play an important role in controlling the tumor microenvironment (TME), and they can control the development, immune escape, angiogenesis, invasion, and migration of certain cancers [17].”
17) Introduction, p2: A paragraph associated to that non-membranous nanoparticles including , albumin, argonaute protein complexes, exomeres lipoproteins, supermeres, supramolecular attack particles, vault and viral particles also carry nucleic acids as well as EVand non-membranous nanoparticles in the text is collectively used instead of “EVs” shall be added after “Exosomes play an important role in controlling the tumor microenvironment (TME), and they can control the development, immune escape, angiogenesis, invasion, and migration of certain cancers [17].”
18) Introduction, p2: Add references to support “In recent years, EVs have been increasingly recognized as important cancer biomarkers because of their high concentration in blood and other body fluids, and because of the bioinformation they carry from their cells of origin in the form of genes and proteins.”
19) Introduction, p2-14: Replace “EVs” by “EVs and non-membranous nanoparticles” in the whole manuscript.
20) Introduction, p2: Add references, and rephrase to support “Non-coding RNA (ncRNA) is a nucleic acid that can be packaged within EVs and transferred among tumor cells.”
21) Introduction, p2: Add references, and rephrase to support “The peculiar structure of EVs, with bilayer membranes, supports ncRNA transmission and protects them from degradation by circulatory nucleases.”
22) Introduction, p2: Most of the findings associated to EVs carrying nucleic acids need to be re-assessed since presence of non-membranous nanoparticles can’t be neglected. Rephrase “In the last years, many reports have shown that EV-related ncRNAs play a pivotal role in many biological and pathological processes [22]. Studies on EV-related ncRNAs derived from HCC cells have been increasing, indicating that further knowledge of EV-related ncRNAs in liver cancer is significant for early diagnosis and treatment.”
23) Introduction, p2: Most of the findings associated to EVs carrying nucleic acids need to be re-assessed since presence of non-membranous nanoparticles can’t be neglected. Rephrase “In this review, the latest studies on various EV-related ncRNAs in HCC will be discussed, with a focus on their potential value as biomarkers of disease as well as in tumor progression, drug resistance, and immune escape (figure 1).”
24) Introduction, p2: Figure 1 content shall be commented in the text to support “In this review, the latest studies on various EV-related ncRNAs in HCC will be discussed, with a focus on their potential value as biomarkers of disease as well as in tumor progression, drug resistance, and immune escape (figure 1).”
25) micro(mi)RNA-EVs,Title, p3: Replace,Title, p2: Replace “EVs” by “Extracellular vesicles and non-membranous nanoparticles” in “. micro(mi)RNA-EVs”
26) micro(mi)RNA-EVs p3: Add references to support “miRNAs are small, non-coding, single-stranded RNA molecules 19-25 nucleotides long which have been implicated in the regulation of post-transcriptional gene expression by silencing messenger RNAs (mRNAs).”
27) micro(mi)RNA-EVs p3: Add references and rephrase “Furthermore, miRNAs contained in HCC-derived EVs are found in biological fluids, so they are also considered cancer biomarker candidates.”
28) micro(mi)RNA-EVs p3: Rephrase “Since different miRNAs are associated with different HCC stages, miRNAs from EVs secreted by HCC tumor cells have an important potential diagnostic and prognostic role [24].”
29) miRNA-EVs and HCC progression, Title, p3: Replace “EVs” by “Extracellular vesicles and non-membranous nanoparticles” and HCC by its full name in the title.
30) miRNA-EVs and HCC progression, p3: It is unclear if such functions are essentially associated to EVs or to EVs and non-membranous nanoparticles. Add references and rephrase “EVs are crucial mediators of both autocrine and paracrine cell communications among different types of liver cells (stellate cells, hepatocytes, and immune cells, including T and B cells, natural killer cells, Kupffer cells) to maintain a physiological state.”
31) miRNA-EVs and HCC progression, p3: Indicate cell types and from which species, how EVs were extracted to support “On the contrary, in a pathological state such as cancer, miRNAs contained in EVs may promote cell growth as in HCC [25].”
32) miRNA-EVs and HCC progression, p3: Indicate specie origin of HCC in “Many miRNAs aberrantly expressed in HCC may act as oncogenes (such as miR-27a) if there is no longer cell cycle control, or may act as tumor suppressors if miRNA blocks the expression of genes involved in cell proliferation (such as miR-9 -3p) [26-27].”
33) miRNA-EVs and HCC progression, p3: Indicate how was extracted exosomal preparation and replace “exosmal” by “EVs and non-membranous nanoparticles” in “Tang J et al. found lower levels of exosomal miR-9-3p in the serum of HCC patients compared to control sera.”
34) miRNA-EVs and HCC progression, p4: I suggest a clear separation between human serum and cell findinds since they shall not be mixed. Move in another section “Conversely, as demonstrated by Cabiati et al. [24], EVs with high levels of miR-27a are more secreted by HCC HepG2 cells than non-tumor hepatocytes.”
35) miRNA-EVs and HCC progression, p4: Indicate specie origin of the HCC cells in “miR-27a promotes HCC cell proliferation through PPAR-γ suppression; in fact, miRNA overexpression is closely related to PPAR-γ down-regulation. PPAR-γ activation arrests the cell cycle through p21 and p53, and it also induces apoptosis activating Fas [26].”
36) miRNA-EVs and HCC progression, p4: Indicate specie origin of the HCC cells, and how were prepared exosomes, replace exosomal by “to EVs and non-membranous nanoparticles” in High exosomal miR-429 expression is involved in HCC progression, targeting Rb binding protein 4 (RBBP4) and activating E2F1, thus promoting POU class 5 homeobox 1(POU5F1) expression [28].”
37) miRNA-EVs and HCC progression, p4: Indicate specie origin of EVs and they were extracted in “EV carried miR-221 targets p27/Kip1 tumor suppressor gene promoting tumor cell proliferation and migration[29].”
38) miRNA-EVs and HCC progression, p4: Move above with the findings associated to human fluids, indicate how EVs were extracted, replace “exosmal” by “EVs and non-membranous nanoparticles” in “The expression levels of exosomal miR-665 in the sera of 30 HCC patients were significantly increased compared with a control group of healthy subjects.
39) miRNA-EVs and HCC progression, p4: Merge with cellular findings, indicate specie origin and how EVs were extracted in “Indeed, Qu et al. demonstrated in vitro that miR-665 contained in HCC-derived EVs promotes cell proliferation and tumor growth through the activation of the MAPK/ERK pathway [30].”
40) miRNA-EVs and HCC progression, p4: Merge with cellular findings, indicate specie origin and how EVs were extracted in “Zhou et al. indicated that HCC-derived exosomal miR-21 could convert normal HSCs (hepatic stellate cells) to CAFs (cancer-associated fibroblasts) by decreasing PTEN expression, leading to the activation of the PDK1/AKT pathway”
41) miRNA-EVs and HCC progression, p4: miRNA-EVs and HCC progression, p4: Indicate specie origin and how EVs were extracted in “EVs and non-membranous nanoparticles” in “miR-93 from HCC-derived EVs promotes HCC proliferation and invasion by directly inhibiting TIMP2/TP53INP1/CDKN1A. Furthermore, the overexpression of miR-93 leads to poor prognosis in HCC patients [32].”
42) Merge with cellular findings, indicate specie origin and how EVs were extracted in “Yang et al. demonstrated that HCC cell-derived EVs with miR-3129 promoted HCC proliferation and epithelial-mesenchymal transition (EMT) in vitro, and they also promoted HCC growth and metastasis formation in vivo [33].”
43) miRNA-EVs and immune escape in HCC, Title, p4: Replace “EVs” by “Extracellular vesicles and non-membranous nanoparticles” and replace “HCC” by its full name in the title.
44) miRNA-EVs and immune escape in HCC, p4: Add references and rephrase in “In addition to tumor progression, HCC-derived EVs modulate many immune cells, attenuating the anti-HCC immune response.”
45) miRNA-EVs and immune escape in HCC, p4: Add references to support “Macrophages play an important role in tumors because they participate in innate immunity. Tumor-associated macrophages (TAMs) consist of two subgroups, M1 and M2.”
46) miRNA-EVs and immune escape in HCC, p4: Add references to support “ “M1 phenotype macrophages expressing pro-inflammatory cytokines inhibit cell proliferation, while the M2 phenotype produces anti-inflammatory cytokines, including TGF-β and IL-10 , and shows immunosuppressive and pro-tumoral effects.”
47) miRNA-EVs and immune escape in HCC, p4: Indicate specie origin, how EVs were extracted, and rephrase “In this context, Liu et al. found that HCC cells transmit exosomal-miR-23a-3p to M2 macrophages, inhibiting PTEN expression and activating the PI3K/AKT signaling pathway; thus, PD-L1 expression increases, and T cell activities are inhibited.”
48) miRNA-EVs and immune escape in HCC, p4: Indicate specie origin, how EVs were extracted, and rephrase “miR-146a-5p, overexpressed in HCC-derived EVs, promotes the M2-like phenotype, causing the downregulation of TNF-α and the differentiation of CD206+ macrophages. [35].”
49) miRNA-EVs and immune escape in HCC, p4: Indicate specie origin, how EVs were extracted, and rephrase “MiR-452-5p secreted by HCC-derived EVs induces the polarization of M2 macrophages, accelerating HCC growth and metastasis by targeting the tumor suppressor TIMP3 [36]”
50) miRNA-EVs and immune escape in HCC, p4: Indicate specie origin, how EVs were extracted, and rephrase “Similarly, HCC-derived exosomal miR-21-5p promotes M2 polarization of macrophages, and it is associated with poor prognosis in HCC [37].”
51) miRNA-EVs and immune escape in HCC, p4: Merge with findings associated to animal cells, indicate how EVs were extracted, and rephrase “Murine HCC H22 cells release EVs loaded with miRNA let-7b that binds macrophages, blocking IL-6 production. Thus, this miRNA attenuates tumor inflammation [38].”
52) miRNA-EVs and immune escape in HCC, p4: Indicate specie origin, how EVs were extracted, and rephrase “miR-92b loaded in HCC-derived EVs from rats has been shown to bind natural killer (NK) cells, causing a decrease of CD69 levels of CD69 and leading to immune escape [39].”
53) miRNA-EVs and HCC drug resistance, Title, p4: Replace “EVs” by “Extracellular vesicles and non-membranous nanoparticles” and replace “HCC” by its full name in the title.
54) miRNA-EVs and immune escape in HCC, p4: Add references to support “Drug resistance represents the primary cause of chemotherapy failure, and it occurs when cancer cells are insensitive to drugs.”
55) miRNA-EVs and immune escape in HCC, p4: Add references to support “Sorafenib is an antineoplastic agent used to treat patients with advanced HCC, and it acts by inhibiting RAF kinases, platelet-derived growth factor receptor (PDGFR-β), vascular endothelial growth factor (VEGFR), and several other kinases.”
56) miRNA-EVs and immune escape in HCC, p4: Add references to support “Moreover, sorafenib may induce p53 expression causing Forkhead box M1 (FOXM1) suppression.”
57) miRNA-EVs and immune escape in HCC, p4: Indicate specie origin, how EVs were isolated, and rephrase to support “miR-25 loaded in HCC-derived EVs mediates sorafenib resistance in HCC by blocking p53, enhancing FOXM1 expression, and activating the HGF/Ras pathway [41].
58) miRNA-EVs and immune escape in HCC, p4: Indicate how EVs were isolated, add reference associated to Wang et al here, and rephrase to support “Wang et al. found that exosome-carried miR-744 l was decreased either in HepG2 cells compared to LO2 cells, and in exosomes from serum of HCC patients compared with those derived from healthy individuals.”
59) miRNA-EVs and immune escape in HCC, p4: Indicate specie origin, how EVs were isolated, add reference and rephrase “HCC cells secrete exosomal miR-1247-3p that decreases B4GALT3, activating the β1- integrin/NF-κB pathway in fibroblasts.”
60) miRNA-EVs and immune escape in HCC, p4: Add more information to support “Activated CAFs increase secretion of IL-6 and IL-8, and other pro-inflammatory cytokines, leading to stemness, EMT, and chemoresistance to sorafenib [43].”
61) miRNA-EVs as biomarkers in HCC, title, p5: Replace “EVs” by “Extracellular vesicles and non-membranous nanoparticles” and replace “HCC” by its full name in the title.
62) miRNA-EVs as biomarkers in HCC, p5: Rephrase “New non-invasive biomarkers, such as EV-loaded miRNAs, have been attracting the attention of the scientific community for diagnosing HCC [44].”
63) miRNA-EVs as biomarkers in HCC, p5: Rephrase, and add references to support “There are several studies on EVs containing non-coding RNA as biomarkers for cancer, but only recently has research on miRNAs as biomarkers of HCC been increasing.
64) miRNA-EVs as biomarkers in HCC, p5: Add references to support “Many miRNAs are recognized for this role, such as miR-224, which is expressed at a significantly higher level in HepG2 and SKHEP1 cells than in healthy liver cells.”
65) miRNA-EVs as biomarkers in HCC, p5: Rephrase, and add references to support “Similarly, serum exosomal miR-224 expression levels are higher in patients with HCC than in healthy controls.”
66) miRNA-EVs as biomarkers in HCC, p5: Indicate how EVs were extracted, and rephrase “Furthermore, high exosomal expression of miR-224 in serum of HCC patients is linked to large and advanced stage tumors. [45].”
67) miRNA-EVs as biomarkers in HCC, p5: Indicate how EVs were extracted, and rephrase miR-21-5p is upregulated in plasma-derived exosomes from patients with HCC compared to patients with liver cirrhosis; instead, miR-92a3p is downregulated. “miR-21-5p is upregulated in plasma-derived exosomes from patients with HCC compared to patients with liver cirrhosis; instead, miR-92a3p is downregulated.”
68) miRNA-EVs as biomarkers in HCC, p5: Indicate how EVs were extracted, add reference here, and rephrase “Cho et al. identified miR-4661-5p in HCC-derived EVs as a reliable biomarker of HCC at all stages.”
69) miRNA-EVs as biomarkers in HCC, p5: Indicate how EVs were extracted, and rephrase “Analyzing the sera of 178 subjects, which included 28 healthy individuals, 27 patients with CHB (chronic hepatitis B), 33 patients with LC (liver cirrhosis) and 90 patients with HCC, exosomal miR-10b-5p was only present at high levels in HCC patients, so it is considered a potential serum biomarker specifically in early-stage HCC [48].”
70) miRNA-EVs as biomarkers in HCC, p5: Indicate how EVs were extracted, specify from which samplrs, and rephrase “Exosomal miR-655, mentioned above, may be considered a new biomarker for HCC diagnosis and prognosis [30].”
71) miRNA-EVs as biomarkers in HCC, p5: Indicate how EVs were extracted, and rephrase “EV-carried miR-3129 is more expressed in the plasma of HCC patients than in healthy subjects, and it is considered a potential new tool for HCC diagnosis [33].”
72) miRNA-EVs as biomarkers in HCC, p5: Indicate how EVs were extracted, add reference here, and rephrase “Boonkawe et al. demonstrated that plasma-derived EV-miR-19-3p was significantly elevated in patients with HCC compared with the healthy individuals.”
73) miRNA-EVs as biomarkers in HCC, p6: Indicate cell origin and from which samples EVs were analyzed to support “High levels of miR-23a are associated with a poor prognosis of HCC, in fact, decreased levels of miR-23a block cell proliferation, as demonstrated by Bao et al. [50]”
74) miRNA-EVs as biomarkers in HCC, p6: : Indicate how EVs were extracted, and rephrase “Cabiati et al. also demonstrated that exosomal miR-23a is increased in HCC patients compared to controls, thus it is used as a biomarker for the diagnosis of early stages of HCC. [24].”
75) miRNA-EVs as biomarkers in HCC, p6: Justify, and add references to support “Finding HCC-derived EV-loaded miRNAs as new biomarkers will improve the performance of HCC surveillance systems in clinical practice.”
76) lncRNA-EVs, Title, p6 : Replace “EVs” by “Extracellular vesicles and non-membranous nanoparticles”
77) lncRNA-EVs, p6 : Add references to support “About 75% of the human genome is actively transcribed, but only 2% of this is represented by sequences encoding proteins; the other sequences left are ncRNAs, and they are RNA that lack protein coding capacity.”
78) lncRNA-EVs, p6 : Add references to support “lncRNAs are important intracellular regulators which can regulate gene expression by interacting with DNA binding promoters orbdistal regulatory elements and recruiting epigenetic modifiers.”
79) lncRNA-EVs, p6 : Add references to support “lncRNAs can regulate mRNA stability by binding it. “
80) lncRNA-EVs, p6 : Add references to support “Besides, lncRNAs can also bind to miRNAs like a molecular sponge, interfering with their activities.”
81) lncRNA-EVs, p6 : Add references to support “lncRNAs have the ability to engage with proteins, contributing to the assembly of protein complexes, where they serve as a scaffold.”
82) lncRNA-EVs and HCC tumor progression, Title, p7: Replace “EVs” by “Extracellular vesicles and non-membranous nanoparticles” and replace “HCC” by its full name in the title.
83) lncRNA-EVs and HCC tumor progression, p7: Justify, add references and rephrase “The analysis of the autocrine effects of the EV contents released by tumor cells has highlighted the role of numerous EV-enriched lncRNAs.”
84) lncRNA-EVs and HCC tumor progression, p7: Add references “lncRNA FAM138B (linc-FAM138B) is downregulated in HCC, and its low expression correlates with a poor prognosis.”
85) lncRNA-EVs and HCC tumor progression, p7: Indicate how EVs are isolated, add references and rephrase linc-FAM138B can be packaged in exosomes and released by HCC cells. HCC cells treated with linc-FAM138B-EVs show reduced proliferation, migration, and invasion due to the inhibition of miR-765 levels.”
86) lncRNA-EVs and HCC tumor progression, p7: Indicate how EVs are isolated, and rephrase “Plasma exosomal lncRNA RP11-85G21.1 (lnc85) stimulated HCC cellular proliferation and migration by targeting miR-324-5p [58].”
87) lncRNA-EVs and HCC tumor progression, p7: The previous findings refered to plasma samples, while the findings here are from cellular origin. Indicate how EVs are isolated, rephrase, and merge the sentence with cell findings in “Exosomal LINC00161 has been recognized as a potential biomarker for HCC, and in HUVEC cells in vitro it was able to stimulate proliferation, migration, and angiogenesis.”
88) lncRNA-EVs and HCC tumor progression, p7: Avoid mixing findings from plasma, cells or from in vivo, indicate animal origin to support “In vivo experiments showed that LINC00161 induces tumorigenesis and metastasis in HCC by targeting miR-590-3p and, consequently, activating its target, ROCK2 [59].”
89) lncRNA-EVs and HCC tumor progression, p7: Indicate specie origin, specify how EVs were isolate, add references and rephrase “lncRNA ASMTL-AS1 is highly expressed in HCC and delivered by exosomes.”
90) lncRNA-EVs and HCC tumor progression, p7: “lncRNA-EVs and HCC tumor progression, p7: Merge with the cell findings “AS-MTL-AS1 enhances cell proliferation, migration, along with promoting invasion and EMT in Huh7 cells via ASMTL-AS1/miR-342-3p/NLK/YAP axis [60].”
91) lncRNA-EVs and HCC tumor progression, p7: Delete “An in vitro study on HCC cells showed that treatment with poprofol downregulates exosome levels of lncRNA H19.”
92) lncRNA-EVs and HCC tumor progression, p7: Indicate how EVs were extracted, specify cell origin, add reference, and rephrase “Treatment with exosome-depleted lncH19 inhibited proliferation, migration, and invasion and induced apoptosis in HCC cells, whereas exosomal lncH19 from untreated HCC cells promoted the proliferation and metastasis, and inhibited apoptosis, of HCC cells.”
93) lncRNA-EVs and HCC tumor progression, p7: Add references to support “Moreover, the Authors found that the overexpression of miR-520a-3p reversed the effects of treatment with H19-Propofol-Huh7-exo”
94) lncRNA-EVs and HCC tumor progression, p7: Add references to support “Special environmental conditions, such as hypoxia, can modulate the release of EVs and define specific cargos.”
95) lncRNA-EVs and HCC tumor progression, p7: Add references to support “This is the case of linc-ROR, which is induced and enriched in exosomes produced by HCC cells under hypoxic stress.”
96) lncRNA-EVs and HCC tumor progression, p7: Delete “Exosomes play an important role in intercellular cross talk that clearly affects tumor progress.”
97) lncRNA-EVs and HCC tumor progression, p7: Indicate how EVs were isolated, indicate specie origin, rephrase, and merge with the cell findings “lncRNA TUG1 presented in CAF-secreted exosomes promoted migration, invasion, and glycolysis in HCC cells [64].”
98) lncRNA-EVs and HCC tumor progression, p7: Merge with animal model findings, add references and rephrase “In a model of DEN-induced HCC in rats, treatment with cancer stem cell (CSC)-derived exosomes upregulated the exosomal lncHEIH, lncHOTAIR, and lncTuc339 in cancer cells.”
99) lncRNA-EVs and HCC tumor progression, p7: Indicate how EVs were isolated, and rephrase in “The expression of these three lncRNAs was downregulated in livers treated with mesenchymal stem cell (MSC) exosomes.”
100) lncRNA-EVs as biomarkers in HCC, Title, p8: Replace “EVs” by “Extracellular vesicles and non-membranous nanoparticles” and replace “HCC” by its full name in the title.
101) lncRNA-EVs as biomarkers in HCC, p8: Justify, and add reviews to support “Numerous exosome-contained lncRNAs have been the subject of extensive research for their considerable diagnostic value as potential and promising diagnostic liquid biopsy markers.”
102) lncRNA-EVs as biomarkers in HCC, p8: It is too broad. Delete “ The diagnostic power of serum EV-lncRNAs for HCC has been shown [67].”
103) lncRNA-EVs as biomarkers in HCC, p8: Indicate how EVs were isolated, from which type of samples, and rephrase “Su et al. identified a predictive signature based on five upregulated exosome-related lncRNAs (AC099850.3, LINC01138, MKLN1-AS, AL031985.3, TMCC1-AS1) associated with poor prognosis [68].”
104) lncRNA-EVs as biomarkers in HCC, p8: Indicate how EVs were isolated, and rephrase The diagnostic power of serum EV-lncRNAs for HCC has been shown [67].
105) lncRNA-EVs as biomarkers in HCC, p8: Indicate how EVs were isolated, and rephrase “Kim et al. identified four serum EV-derived lncRNAs, SNHG1, MALAT1, HOTTIP, and DLEU2, which were differentially expressed and significantly discriminated between the HCC and non-HCC samples [69]”
106) lncRNA-EVs as biomarkers in HCC, p8: Indicate how EVs were isolated, and rephrase “Exosomal LINC00161 has been found upregulated in serum exosomes of HCC patients and as suggested by the Authors, can be a valuable marker for the diagnosis of HCC [70].”
107) lncRNA-EVs as biomarkers in HCC, p8: Add references, and explain the contradiction to support “On the contrary, the exosomal lncSENP3-EIF4A1 exhibits reduced expression in HCC patients compared to healthy controls.”
108) lncRNA-EVs as biomarkers in HCC, p8: Add references, indicate how EVs were extracted, and rephrase “Serum exosomal lncRNA CRNDE expression levels were shown to be higher in patients with HCC compared to normal controls. CRNDE overexpression was associated with cell proliferation, migration, and invasion of HCC cells.”
109) lncRNA-EVs as biomarkers in HCC, p8: Indicate how EVs were extracted, and rephrase “In addition, high levels of serum exosomal lncRNA CRNDE were reported to be correlated with poor prognosis in HCC patients [72].”
110) lncRNA-EVs as biomarkers in HCC, p8: Add references, indicate how EVs were extracted, and rephrase “Recently, Yao et al. demonstrated that, compared to normal controls, plasmatic exosomal PRKACA-202, H19-204, and THEMIS2-211 were upregulated in HCC patients. In this study, the diagnostic value of the combination of exosomal THEMIS2-211 and PRKACA-202 surpassed that of AFP for the diagnosis of early-stage HCC patients.”
111) lncRNA-EVs as biomarkers in HCC, p8: Add references, indicate how EVs were extracted, and rephrase “Exosome-derived lncRNAs CTD-2116N20.1 and RP11-136I14.5, have been recognized as potential biomarkers for predicting the survival rate of HCC patients. Their presence is associated with an unfavorable prognosis in individuals with HCC.”
112) lncRNA-EVs as biomarkers in HCC, p8: Add references, indicate how EVs were extracted, and rephrase “The exosomal lncRNA RP11-583F2.2 was first individuated by bioinformatic analysis and then validated in the serum of HCC patients. Its expression is upregulated in HCC patients compared to viral hepatitis C patients and healthy people.”
113) lncRNA-EVs as biomarkers in HCC, p8: Indicate how EVs were extracted, and rephrase “Similarly, exosomal lncRNA-HEIH expression levels allowed researchers to distinguish between chronic hepatitis C and hepatitis C virus (HCV)-associated HCC patients as its expression increased only in exosomes from patients with HCV-related HCC [76].”
114) lncRNA-EVs as biomarkers in HCC, p8: Indicate how EVs were extracted, and rephrase “Plasma exosomal lncRNA RP11-85G21.1 (lnc85) allowed AFP-negative HCC to be distinguished from liver cirrhosis patients and healthy controls [58].”
115) lncRNA-EVs as biomarkers in HCC, p8: Indicate how EVs were extracted, and rephrase “The exosomal lncRNAs ENST00000440688.1, ENST00000457302.2, and ENSG00000248932.1 are differentially expressed in patients with metastatic HCC compared to those with non-metastatic HCC [77]. “
116) lncRNA-EVs as biomarkers in HCC, p8: Indicate how EVs were extracted, and rephrase “LINC00635 and lncRNA ENSG00000258332.1 (LINC02394) were upregulated in HCC patients and combined with AFP, were able to distinguish HCC patients in an independent test of validation [78].”
117) lncRNA-EVs as biomarkers in HCC, p9: Indicate how EVs were extracted, and rephrase “The expression of serum exosomal lncRNA-ATB, associated with exosomal miRNA-21, was inversely correlated with the overall survival and progression-free survival of HCC patients [79].”
118) lncRNA-EVs as biomarkers in HCC, p9: Add reference, indicate how EVs were extracted, and rephrase “Analysis of EVs derived from the serum of healthy individuals and individuals with hepatitis, cirrhosis, and HCC revealed differential expression of lncZEB2-19, lnc-GPR89B-15, lnc-EPC1-4, and lnc-FAM72D-3 in HCC patients.”
119) lncRNA-EVs and immunomodulatory capacity, Title, p9: Replace “EVs” by “Extracellular vesicles and non-membranous nanoparticles” in the title.
120) lncRNA-EVs and immunomodulatory capacity, p9: Add reference, indicate how EVs were extracted, from which samples, and rephrase “HCC derived exosomes were found to be enriched with lncTUC339, which is involved in modulating tumor cell growth and adhesion.”
121) lncRNA-EVs and immunomodulatory capacity, p9: Add reference, indicate how EVs were extracted, from which cell specie, and rephrase “It was demonstrated that exosomes from HCC cells could be taken up by THP-1 cells where TUC339 induces macrophage M1/M2 polarization, the switch from a proinflammatory (M1) to an anti-inflammatory phenotype (M2).
122) lncRNA-EVs and immunomodulatory capacity, p9: Add reference to support “In addition, elevated levels of TUC339 were found in M(IL-4) macrophages.”
123) lncRNA-EVs and immunomodulatory capacity, p9: Add references to support ”The immunomodulatory capacity of the HCC released exosomal lncRNA has been demonstrated by other studies.”
124) lncRNA-EVs and immunomodulatory capacity, p9: Specify PD, and add references in “In HCC, the overexpression of PD ligands was linked with poor prognosis in HCC.”
125) lncRNA-EVs and immunomodulatory capacity, p9: Indicate how EVs were extracted, indicate cell specie, and rephrase “Fan et al. demonstrated that HCC cells released lncRNA PCED1B-AS1 containing exosomes that enhanced PD-L expression in receipt HCC cells. PCED1B-AS1 induced PDL expression via sponging hsa-mir-194-5p, which inhibited PD-L expression [81].”
126) lncRNA-EVs and drug resistance in HCC, Title, p9: Replace “EVs” by “Extracellular vesicles and non-membranous nanoparticles” and replace “HCC” by its full name in the title.
127) lncRNA-EVs and drug resistance in HCC, p9: Indicate how EVs were extracted, indicate cell specie, and rephrase “lncRNA-ROR was found to be enriched in HCC-derived exosomes and seemed to be involved in mechanisms of sorafenib resistance; indeed, lncRNA-ROR levels increased during sorafenib treatment and inhibited sorafenib-induced cell death [82].”
128) circular (circ) RNA-EVs, Title, p10 : Replace “EVs” by “Extracellular vesicles and non-membranous nanoparticles” in the title.
129) circular (circ) RNA-EVs, p10 : Add references to support “Circular RNAs (circRNAs) are endogenous non-coding products.”
130) circRNA-EVs and HCC tumor progression, Title, p11: Replace “EVs” by “Extracellular vesicles and non-membranous nanoparticles” and replace “HCC” by its full name in the title.
131) circRNA-EVs and HCC tumor progression, p11: “Indicate how EVs were extracted, indicate cell specie, and rephrase “Numerous studies have found a close relationship between the presence of exosomal circRNAs and HCC progression, invasion, angiogenesis, metastasis, and EMT [91].”
132) circRNA-EVs and HCC tumor progression, p11: Indicate how EVs were extracted, and rephrase “Dai et al. demonstrated that L-02 cells exposed to arsenite release circRNA_100284 contained in exosomes which is able to regulate the cell cycle and proliferation of normal liver cells by interacting with miR-217 [92]”
133) circRNA-EVs and HCC tumor progression, p11: Indicate how EVs were extracted, indicate cell specie, and rephrase “Exosomal circRNAs also mediate the cross-talk between normal and cancer cells to regulate HCC growth and metastasis [96].”
134) circRNA-EVs and HCC tumor progression, p11: Add reference to support “EMT is a fundamental biological process for cancer cell invasion.”
135) circRNA-EVs and HCC tumor progression, p11: Indicate how EVs were extracted, indicate cell specie, and rephrase “Zhang and colleagues showed that circ_0003028, through an exosome pathway, controls E-cadherin, N-cadherin, and vimentin [97]”
136) circRNA-EVs and HCC tumor progression, p11: Indicate how EVs were extracted, indicate cell specie, and rephrase “The exosome circWDR25 induced HCC progression and EMT through ALOX15 activation by miR-4474-3p-sponge [98].”
137) circRNA-EVs and HCC tumor progression, p11: Indicate how EVs were extracted, indicate types of in vitro and in vivo findings, and rephrase “In vivo and in vitro studies indicated that in HCC cells the high levels of exosomal circ-0004277 induces growth, invasiveness and EMT [99].”
138) circRNA-EVs and HCC tumor progression, p11: Indicate how EVs were extracted, add reference, and rephrase “High levels of circ_002136 have shown to be contained into exosomes from HCC cell 4lines (HA22T and Huh7).”
139) circRNA-EVs and HCC tumor progression, p11: Indicate how EVs were extracted, indicate types of in vitro and in vivo findings, and rephrase “In vitro and in vivo experiments showed that silencing circ_002136 inhibited the growth of HCC cells via the miR-19a-3p and RAB1A axis [100].”
140) circRNA-EVs and immune escape, Title, p11: Replace “EVs” by “Extracellular vesicles in the title.
141) circRNA-EVs and immune escape, p11: It was repeated in the following sentences. Delete “Numerous studies have demonstrated that exosomal circRNAs derived from cancer cells can take part in the tumor immune escape program.”
142) circRNA-EVs and immune escape, p11: Indicate how EVs were extracted, indicate cell specie, add reference, and rephrase “Wang et al. have shown that HCC cells transfer circ_0074854 in macrophages through exosomes.”
143) circRNA-EVs and immune escape, p11: Indicate how EVs were extracted, and rephrase “Recently, Hu and colleagues reported that conditioned medium from cultured HCC cells and plasma-exosomes from HCC patients showed high levels of circCCAR1 [103].”
144) circRNA-EVs and immune escape, p12: Indicate how EVs were extracted, indicate cell specie, and rephrase “A recent study found that HCC cell-derived exosomal circGSE1 promoted the tumor immune escape process by expanding T cells (Tregs) via regulating the miR-324- 5p/TGFBR1/Smad3 axis. circGSE1 acted as a sponge for miR-324-5p. As a result, the expansion of Tregs promoted the progression of HCC [106].”
145) circRNA-EVs and immune escape, p12: Indicate how EVs were extracted, indicate cell specie, add reference, and rephrase “It was demonstrated that macrophages absorb exosome circMEM181, leading in the regulation of CD39 expression through sponging miR-488-3p.It.”
146) circRNA-EVs and drug resistance in HCC, Title, p12: Replace “EVs” by “Extracellular vesicles and non-membranous nanoparticles” and replace “HCC” by its full name in the title.
147) circRNA-EVs and drug resistance in HCC, p12: Indicate how EVs were extracted, indicate cell specie, add reference, and rephrase “In HCC the cell-derived exosomal circRNA-SORE, also named circ_0087293 or circRNA_104797, controlled sorafenib resistance by stabilizing the oncoprotein YBX1.”
148) circRNA-EVs and drug resistance in HCC, p12: Delete “Exosomal circPAK1 has been demonstrated to be involved in transmitting lenvatinib resistance.”
149) circRNA-EVs and drug resistance in HCC, p12: Indicate how EVs were extracted, indicate cell specie, and rephrase “Hao et al. demonstrated that circPAK1 was overexpressed in lenvatinib resistant cell lines, and exosomes from resistant cells can mediate circPAK1 transfer to sensitive cells inducing the lenvatinib resistance of receipt cells [112].”
150) circRNA-EVs and drug resistance in HCC, p12: Add references to support “Other Authors have demonstrated that circZFR was highly expressed in cisplatin resistant HCC cell lines and in CAFs.”
151) circRNA-EVs and drug resistance in HCC, p12: Indicate how EVs were extracted, indicate cell specie, and rephrase “Furthermore, CAF-derived exosomes delivered circZFR to HCC cells, inhibited the STAT3/NF-κB axis, and induced cisplatin resistance [113].”
152) circRNA-EVs as biomarkers in HCC, Title, p12: Replace “EVs” by “Extracellular vesicles and non-membranous nanoparticles” and replace “HCC” by its full name in the title.
153) circRNA-EVs as biomarkers in HCC, p12: It was repeated below. Delete “Recently, several studies have shown that exosomal circRNAs function as new biomarkers for HCC.”
154) circRNA-EVs as biomarkers in HCC, p12: Delete “Exosomal circRNAs have been detected in human body fluids and have been used as noninvasive biomarkers for HCC tumor detection.”
155) circRNA-EVs as biomarkers in HCC, p12: Indicate how EVs were extracted, indicate types of tumor tissues, and rephrase “Other relevant evidence comes from the work of Lin et al. who found that circ_0072088 was highly expressed in tumor tissues as well as in exosomes isolated from serum of patients with HCC.”
156) circRNA-EVs as biomarkers in HCC, p12: Delete “Similarly, circFBLIM1 was shown to be upregulated in HCC serum exosomes.”
157) circRNA-EVs as biomarkers in HCC, p12: Indicate samples, and rephrase “CircFBLIM1 was shown to function as miR-338 sponge, and its levels were correlated with HCC glycolysis and progression induced by the circFBLIM1/miR-338/LRP6 axis [116].”
158) circRNA-EVs as biomarkers in HCC, p12: Indicate how EVs were extracted, and rephrase “Serum exosomal hsa_circ_0028861 and hsa_circ_0070396 were detected at higher levels in HCC patients compared to chronic HBV and cirrhosis individuals [117-118].”
159) circRNA-EVs as biomarkers in HCC, p13: It was unclear if EVs were analyzed in “In addition, Zhang et al. found that in HCC samples and adjacent samples collected from 68 HCC patients, levels of circTMEM45A were higher in the HCC samples, and its level was positively correlated with poor prognosis in patients.”
160) circRNA-EVs as biomarkers in HCC, p13: Indicate how EVs were extracted, and rephrase “Recently, circANTXR1 was found at high levels in exosomes from serum of HCC patients and exhibited diagnostic value to distinguish HCC patients from healthy controls.”
161) Conclusions, p14: It was repeated in the next sentence. Delete “Many studies have confirmed the significant role played by exosomes in the pathophysiological processes of HCC. ”
162) Conclusions, p14: It was not clear if the effects associated to EVs were solely due to EVs or to other non-membranous nanoparticles. In this respect nanoparticles can be adsorbed as corona particles on the surface of EVs, which is an important issue not discussed. Rephrase “Exosomes and their ncRNA content have an autocrine action modulating cell proliferation and tumorigenicity of the same cell that releases them; they have a paracrine action that conditions the entire tumor environment, modulating the behavior of other cell types, such as stem and immune cells.”
163) Conclusions, p14: Delete “In summary, this study confirms that EV-related ncRNAs are an extremely valuable and revolutionary approach and tool for precision medicine.”
Reviewer 3 Report
Comments and Suggestions for Authors
Brief summary:
In this review, the authors discussed the diverse roles of the non-coding RNA (including micro RNAs, long non‑coding RNAs, and circular RNAs) of HCC-derived extracellular vesicles. These non-coding RNA might play primary roles in the tumor development and progression, in the regulation of tumor microenvironment (TME) and immune escape and drug resistance of HCC.
1. What is the main question addressed by the research?
The authors want to address the roles of non-coding RNA in the HCC.
2. Do you consider the topic original or relevant in the field? Does it address a specific gap in the field?
The topic is relevant in the field. However, it is not novel or original.
3. What does it add to the subject area compared with other published material?
This is a review article. It does not add anything new to the topic. However, the article could provide a comprehensive review on the role of non-coding RNA in the HCC.
4. What specific improvements should the authors consider regarding the methodology? What further controls should be considered?
The authors can add a paragraph to discuss the prospective or future directions on non-coding RNA. Till now, the non-coding RNA is still far from routine clinical use. What kinds of effort should we do?
5. Are the conclusions consistent with the evidence and arguments presented and do they address the main question posed?
The conclusions are consistent with the arguments presented.
6. Are the references appropriate?
The references are appropriate.
7. Please include any additional comments on the tables and figures.
No additional comments on the tables and figures.
Round 2
Reviewer 2 Report
Comments and Suggestions for Authors
General comments:
I) This is ambitious attempt to review the nanoparticle-related non-coding RNAs in heptacellular carcinoma. However, the review lacked a critical assessment on the findings to grasp the diversity of nanoparticles carrying nucleic acids and their possible effects.
The presence of nanoparticles carrying non-coding RNA, in both biological fluids and cell culture media, in addition to extracellular vesicles, is a contested and important topic which obviously needs to be taken into consideration, not only in hepatocellular carcinoma. In this review, our aim was to take into consideration literature referring to extracellular vesicles/exosomes. However, and as suggested by the reviewer, we have modified the text by adding an evaluation of this issue.
Reviewer’s answer: It was partially addressed with regard to point II)
II) One fundamental issue which was not taken sufficiently into account in the review is that extracellular vesicles may not represent the best carriers of nucleic acids. Indeed, several non-membranous nanoparticles, as albumin, argonaute protein complexes, exomeres lipoproteins, supermeres, supramolecular attack particles, vault and viral particles, contain an enriched amount of nucleic acids. Most of the findings associated to extracellular vesicles and nucleic acids need to be re-assessed since the presence of any non-membranous nanoparticles in the samples containing extracellular vesicles can’t be neglected.
This is a very important concept that obviously needs to be stressed. Accordingly, we have modified the text, adding assessment of this issue.
Reviewer’s answer: A paragraph was addressed to underline that extracellular vesicles may not represent the best carriers of nucleic acids, which addresses well the point. However, this concerns shall be reflected in all the manuscript. I suggest that “nanoparticles” rather “extracellular vesicles “ shall be used in the whole text.
Minor comments:
1) Title, p1: Replace “Extracellular vesicle” by “Extracellular vesicles and non-membranous nanoparticles”
Although the reviewer suggested adding the term "non-membranous particles" to the titles of the various subparagraphs, the term "non-membranous nanoparticles" was not introduced as requested in points 1-2-3-4-5-6-7-19-25-29-33-36-38-41-43-53-61-76-82-100-119-126-128-130-146-152.
We are aware of the presence of “non-membranous particles” in exosome preparations, and this issue has also now been highlighted in the introduction of the revised version. However, we would like to highlight the fact that in none of the publications we have cited is there any way to know what type of “non-membranous particles” were present in the purified exosomes used in the various studies. All the peer-reviewed articles cited in our review simply report the term “extracellular vesicles” or “exosomes”, without any reference to the possible presence/absence of “non-membranous particles”. For this reason, we believe that we cannot add the term "non-membranous nanoparticles" in the titles as the reader will not find any information about them in the various paragraphs.
Reviewer’s answer: It was insufficiently addressed. The title is misleading since it suggests that non-coding RNA are found in extracellular vesicles, while it is almost certain that all extracellular vesicles, as reported so far, contain nonvesicular nanoparticles since there were no further experimental evidence to support that extracellular vesicles were devoid of nonvesicular nanoparticles. Most of the findings claiming that extracellular vesicles are carrying non-coding RNA need to be re-assessed. I agree that my previous suggestion “Extracellular vesicles and non-membranous nanoparticles” is not adequate. I suggest to replace “Extracellular vesicle” by “Nanoparticle”.
3) Simple summary, p1: Replace “Extracellular vesicles” by “Extracellular vesicles and non-membranous nanoparticles” in “Extracellular vesicles exert their biological functions by delivering different biomolecules, including non-coding RNAs.”
See previous comment (point 1)
Reviewer’s answer. I suggest to replace “Extracellular vesicle” by “Nanoparticle”.
3) Simple summary, p1: Replace “Extracellular vesicles” by “Extracellular vesicles and non-membranous nanoparticles” in “In this review, the diverse roles of the non-coding RNA cargo of hepatocellular carcinoma-derived extracellular vesicles will be discussed.”
See previous comment (point 1)
Reviewer’s answer. I suggest to replace “Extracellular vesicle” by “Nanoparticle”.
4) Abstract, p1: Add “Non-membranous nanoparticles are also carriers of nucleic acids. Since their presence in the extracellular vesicles can’t be neglected. Extracellular vesicles and nanoparticles shall be considered in the review” or something similar after “Extracellular vesicles (EVs) are nanosized lipid double-layer vesicles containing various biomolecule cargoes, such as lipids, proteins, and nucleic acids.”
See previous comment (point 1)
Reviewer’s answer. I suggest to replace “Extracellular vesicle” by “Nanoparticle”.
5) Abstract, p1: Replace “tumor cell-derived EVs” by “tumor cell-derived EVs and non-membranous nanoparticles” in “Moreover, it has been observed that non-coding RNAs (ncRNAs) carried by tumor cell-derived EVs promote tumorigenesis, regulating the tumor microenvironment (TME) and playing critical roles in the progression, angiogenesis, metastasis, immune escape and drug resistance of HCC.”
See previous comment (point 1)
Reviewer’s answer. I suggest to replace “Extracellular vesicle” by “Nanoparticle”.
6) Abstract, p1: Replace “EV-related ncRNAs” by “EV and non-membranous nanoparticle-related ncRNAs” in “EV-related ncRNAs can provide information on disease status, thus covering a role as biomarkers.”
See previous comment (point 1)
Reviewer’s answer. I suggest to replace “Extracellular vesicle” by “Nanoparticle”.
7) Abstract, p1: Replace “HCC-derived EVs”by “HCC-derived EVs and non-membranous nanoparticles” in “In this review, we discuss the main roles of ncRNAs present in HCC-derived EVs, including micro(mi) RNAs, long non-coding (lnc) RNAs, and circular (circ) RNAs, and their potential clinical value as biomarkers and therapeutic targets.”
See previous comment (point 1)
Reviewer’s answer. I suggest to replace “Extracellular vesicle” by “Nanoparticle”.
16) Introduction, p2: A paragraph associated to the difficulty of obtaining a pure fraction EVs and that non-membranous nanoparticles can’t be excluded from any preparation of EVs shall be added after “Exosomes play an important role in controlling the tumor microenvironment (TME), and they can control the development, immune escape, angiogenesis, invasion, and migration of certain cancers [17].”
We thank the reviewer for this suggestion: we have added a section in the introduction which highlights the difficulty of obtaining a pure EV fraction free of contamination.
Reviewer’s answer: It was addressed but a reference is needed to support “The isolation of EVs is a critical process, with the chosen method significantly impacting both sample yield and purity.”
Another reference is needed to support “After isolation, the second step in studying of EVs is their characterization in terms of physical and biochemical characteristics, as well as content. “
Another reference is needed to support “The most common approach to defining the physicochemical and molecular characteristics of EVs is single par- ticle analysis. “
Another reference is needed to support “Such basic methods as protein quantification, Western Blot, qPCR, omics analysis including proteomics, lipidomic analysis, and RNA sequencing are the main methodologies used to characterize EV cargoes.”
19) Introduction, p2-14: Replace “EVs” by “EVs and non-membranous nanoparticles” in the whole manuscript.
See previous comment (point 1)
Reviewer’s comment: It was not addressed. After the sentence “Non-coding RNA (ncRNA) is a nucleic acid that can be packaged within EVs and transferred among tumor cells [54].” I suggest to explain that most papers which report that non-coding RNA (ncRNA) can be packaged within EVs, did not assess the contribution of non-vesicular particles carrying non-coding RNA. Instead of EVs, the general term of nanoparticles (which comprise EVs) shall be used in the text.
23) Introduction, p2: Most of the findings associated to EVs carrying nucleic acids need to be re-assessed since presence of non-membranous nanoparticles can’t be neglected. Rephrase “In this review, the latest studies on various EV-related ncRNAs in HCC will be discussed, with a focus on their potential value as biomarkers of disease as well as in tumor progression, drug resistance, and immune escape (figure 1).”
See previous comment (point 1)
Reviewer’s comment: See point 19
162) Conclusions, p14: It was not clear if the effects associated to EVs were solely due to EVs or to other non-membranous nanoparticles. In this respect nanoparticles can be adsorbed as corona particles on the surface of EVs, which is an important issue not discussed. Rephrase “Exosomes and their ncRNA content have an autocrine action modulating cell proliferation and tumorigenicity of the same cell that releases them; they have a paracrine action that conditions the entire tumor environment, modulating the behavior of other cell types, such as stem and immune cells.”
See previous comment (point 1) regarding non-membranous nanoparticles. Furthermore, information on corona particles has been added in the introduction. We have replaced Exosomes with EVs.
Reviewer’s comment: See point 19
Author Response
Although the reviewer suggested replacing the term “Extracellular vesicle” by “Nanoparticle”, the term "non-membranous nanoparticles" was not introduced as requested in points I, II, 1-7, 19, 23, 162.
16) Introduction, p2: A paragraph associated to the difficulty of obtaining a pure fraction EVs and that non-membranous nanoparticles can’t be excluded from any preparation of EVs shall be added after “Exosomes play an important role in controlling the tumor microenvironment (TME), and they can control the development, immune escape, angiogenesis, invasion, and migration of certain cancers [17].”
We thank the reviewer for this suggestion: we have added a section in the introduction which highlights the difficulty of obtaining a pure EV fraction free of contamination.
Reviewer’s answer: It was addressed but a reference is needed to support “The isolation of EVs is a critical process, with the chosen method significantly impacting both sample yield and purity.”
Another reference is needed to support “After isolation, the second step in studying of EVs is their characterization in terms of physical and biochemical characteristics, as well as content. “
Another reference is needed to support “The most common approach to defining the physicochemical and molecular characteristics of EVs is single par- ticle analysis. “
Another reference is needed to support “Such basic methods as protein quantification, Western Blot, qPCR, omics analysis including proteomics, lipidomic analysis, and RNA sequencing are the main methodologies used to characterize EV cargoes.”
References have been added.